# The Burer-Monteiro SDP method can fail even above the Barvinok-Pataki bound

**Liam O'Carroll**
Department of Computer Science
Northwestern University
Evanston, IL 60208
liamocarroll2023@u.northwestern.edu

**Vaidehi Srinivas**
Department of Computer Science
Northwestern University
Evanston, IL 60208
vaidehi@u.northwestern.edu

**Aravindan Vijayaraghavan**
Department of Computer Science
Northwestern University
Evanston, IL 60208
aravindv@northwestern.edu

## Abstract

The most widely used technique for solving large-scale semidefinite programs (SDPs) in practice is the non-convex Burer-Monteiro method, which explicitly maintains a low-rank SDP solution for memory efficiency. There has been much recent interest in obtaining a better theoretical understanding of the Burer-Monteiro method. When the maximum allowed rank $p$ of the SDP solution is above the Barvinok-Pataki bound (where a globally optimal solution of rank at most $p$ is guaranteed to exist), a recent line of work established convergence to a global optimum for generic or smoothed instances of the problem. However, it was open whether there even exists an instance in this regime where the Burer-Monteiro method fails. We prove that the Burer-Monteiro method can fail for the Max-Cut SDP on $n$ vertices when the rank is above the Barvinok-Pataki bound ($p \geq \sqrt{2n}$). We provide a family of instances that have spurious local minima even when the rank $p = n/2$. Combined with existing guarantees, this settles the question of the existence of spurious local minima for the Max-Cut formulation in all ranges of the rank and justifies the use of beyond worst-case paradigms like smoothed analysis to obtain guarantees for the Burer-Monteiro method.

## 1 Introduction

Semidefinite programs (SDPs) are a powerful algorithmic tool with wide-ranging applications in combinatorial optimization, control theory, machine learning and operations research. Notably, they yield optimal approximation algorithms for NP-hard problems like Max-Cut [GW95] and other constraint satisfaction problems [Rag08]. While interior-point algorithms and the ellipsoid method give polynomial time guarantees for solving semidefinite programs, memory becomes a bottleneck for relatively modest instance sizes. This has prompted research into scalable semidefinite programming algorithms—see [MHA20] for a recent survey.

One of the most popular methods for solving SDPs in practice is the one pioneered by Burer and Monteiro [BM03, BM05], which explicitly constrains the rank of the SDP solution for efficiency.

36th Conference on Neural Information Processing Systems (NeurIPS 2022).

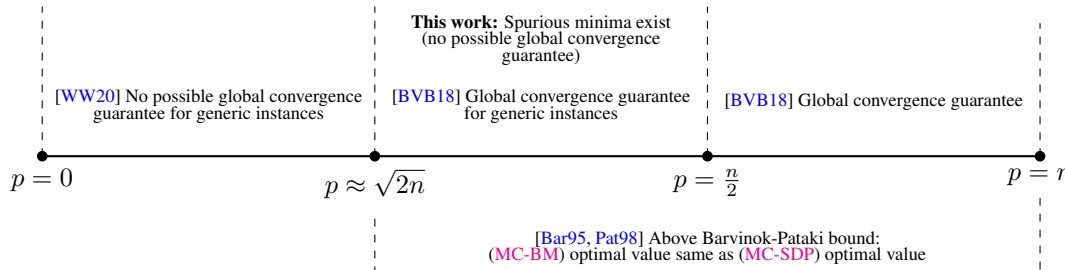

Figure 1: Combined with existing guarantees, our work settles the question of the existence of spurious local minima for (MC-BM) in all ranges of the rank $p$.

Consider the celebrated Goemans-Williamson SDP relaxation for Max-Cut [GW95]:

$$\min_{X \in \mathbb{S}^{n \times n}} \langle A, X \rangle$$
$$\text{s.t.} \quad X_{ii} = 1 \text{ for } i \in [n], \tag{MC-SDP}$$
$$X \succeq 0.$$

Here $\mathbb{S}^{n \times n}$ denotes the set of all symmetric $n \times n$ matrices and the cost matrix $A \in \mathbb{S}^{n \times n}$ is typically the adjacency matrix of a weighted graph.[1] The above Goemans-Williamson SDP also gives a natural semidefinite programming relaxation for the Grothendieck problem [AN04], quadratic programming [Nes98, CW04], variants of community detection problems [Abb17], and other combinatorial optimization problems; in many of these settings the matrix $A$ may also have negative entries. Instead of maintaining a solution $X$ in $\mathbb{S}^{n \times n}$, the Burer-Monteiro method explicitly maintains a low-rank solution of the form $X = YY^\top$ where $Y \in \mathbb{R}^{n \times p}$, and aims to solve the following optimization problem:

$$\min_{Y \in \mathbb{R}^{n \times p}} \langle A, YY^\top \rangle$$
$$\text{s.t.} \quad ||Y_i||^2 = 1 \text{ for } i \in [n], \tag{MC-BM}$$

where $Y_i \in \mathbb{R}^p$ denotes the $i$th row of $Y$, taken as a column vector, and $|| \cdot ||$ denotes the Euclidean norm. We may also denote the objective of (MC-BM) as $\text{OBJ}(Y) := \langle A, YY^\top \rangle$. We denote the feasible region as

$$\mathcal{M}_{n,p} := \left\{ Y \in \mathbb{R}^{n \times p} : ||Y_i||^2 = 1 \text{ for } i \in [n] \right\}.$$

(When $n$ is clear from the context or unimportant, we may just write $\mathcal{M}_p$.) Note that $\mathcal{M}_{n,p}$ is a product manifold since it can be viewed as a Cartesian product of $n$ unit spheres in $\mathbb{R}^p$.

This formulation yields significant memory savings when $p \ll n$ by storing $Y$ instead of $X$ and has the additional advantage of dropping the positive semidefiniteness constraint. On the other hand, (MC-BM) is a non-convex constrained optimization problem. Local optimization methods like Riemannian gradient descent and other heuristics for manifold optimization or constrained optimization are used to solve this non-convex problem with surprisingly good empirical results [BM03, BM05]. This motivates the following question:

*When does the Burer-Monteiro method converge to a globally optimal solution?*

This question has attracted much recent interest on the theoretical front. It is known that when the rank bound $p$ of the SDP solution is above the so-called *Barvinok-Pataki bound* $p \geq \sqrt{2n}$ (more formally, $p$ satisfies $\frac{p(p+1)}{2} \geq n$), a globally optimal solution of rank at most $p$ is guaranteed to exist [Bar95, Pat98]. Moreover, below this bound, there exist instances for which (MC-BM) has a spurious local minimum [WW20]. When the rank bound $p$ is above the Barvinok-Pataki bound, Boumal, Voroninski, and Bandeira [BVB16, BVB18] showed that for *generic* instances,[2] any second-order critical point $Y$ of (MC-BM) is globally optimal for (MC-BM), and thus $YY^\top$ is optimal for

---

[1]Note that (MC-SDP) can also be reformulated as a maximization SDP where the cost matrix is the Laplacian matrix $L = D - A$, where $D$ is the diagonal degree matrix. The two formulations are equivalent since changing the diagonal of the cost matrix just corresponds to adding a constant to the objective value at each feasible point.

[2]By generic instances, we mean the guarantee holds for all cost matrices $A \in \mathbb{S}^{n \times n}$ except a set of zero measure.

(MC-SDP) (their result also applies for a broad class of SDPs with equality constraints). Under such conditions, known algorithms converge to second-order critical points of (MC-BM), and polynomial-time convergence guarantees can be shown for smoothed instances [BBJN18, PJB18, CM19].

On the other hand, for general cost matrices $A$, the best known bound only guarantees convergence to a global optimum of the Burer-Monteiro method when $p > \frac{n}{2}$ [BVB18, Cor. 5.11]. For general cost matrices $A$ of (MC-BM), there is a large range for the rank bound $p \in (\sqrt{2n}, n/2)$ where we do not know whether the Burer-Monteiro method works. To the best of our knowledge, it was open whether there exists *any* instance (not specific to Max-Cut) where the Burer-Monteiro method fails above the Barvinok-Pataki bound. In fact, the authors of [BVB18] pose this question:

> *"It remains unclear whether or not a zero-measure set of cost matrices must be excluded. Resolving this question is key to gaining a deeper understanding of the relationship between* (MC-SDP) *and* (MC-BM)*."*
>
> —End of Section 6 in [BVB18], mutatis mutandis

Our main theorem resolves this question by constructing an instance of (MC-BM) with a spurious local minimum (i.e., a local minimum which is not globally optimal) when $p$ is as large as $\Theta(n)$ (note that a spurious local minimum is also a spurious second order critical point). This result is contextualized in Figure 1.

**Theorem 1.** *For any $n \geq 4$ and $2 \leq p \leq n/2$, there exist cost matrices $A$ for which the associated instance of* (MC-BM) *has a spurious local minimum.*

Combining Theorem 1 with other existing results yields a clearer understanding of the optimization landscape and a characterization of the range of $p$ for which (MC-BM) can have spurious local minima. Furthermore, our result justifies the use of beyond worst-case paradigms like smoothed analysis to obtain global convergence guarantees for the Burer-Monteiro method. (See also Appendix A for a further discussion of prior work.)

We also empirically evaluate Theorem 1 in Appendix G. Our experiments suggest that the spurious local minima we construct have significantly larger basins of convergence than our theoretical results guarantee.

**Outline of paper.** We begin with Section 2 (Preliminaries) which provides necessary background for the construction of spurious local minima in Theorem 1. In Section 3, we list the main lemmas for Theorem 1 and show how Theorem 1 follows. In Sections 4 and 5, we discuss some of these lemmas in more detail. In particular, Section 4 sketches the proofs that our construction yields a spurious first and second-order critical point. Section 5 contains the full proof that our construction yields a spurious local minimum, with the proofs of key sublemmas left to Appendix D. We present potential follow-up directions in Section 6 (Conclusion) and empirically evaluate Theorem 1 in Appendix G.

## 2  Preliminaries

Section 2.1 gives an overview of the Riemannian geometry of (MC-BM). In Section 2.2, we use this geometry to give necessary conditions for local minimality and a characterization of global optimality. Section 2.3 provides an overview of Riemannian gradient descent, which is key to proving local minimality in Theorem 1 (see Section 5 for details). Section 2.4 contains definitions (including classes of matrices) used in the construction for Theorem 1.

### 2.1  Riemannian derivatives

$\mathcal{M}_p$ is a smooth embedded submanifold of $\mathbb{R}^{n \times p}$ [BVB18, Prop. 1.2], and is furthermore an embedded Riemannian submanifold of $\mathbb{R}^{n \times p}$ [Bou22, Def. 3.55] if we equip the linearizations of $\mathcal{M}_p$ at each point (known as the *tangent spaces*—defined below) with (a restriction of) the inner product $\langle \cdot, \cdot \rangle$ on $\mathbb{R}^{n \times p}$.

**Proposition 1** (Tangent space [BVB18, Lem. 2.1]). *The tangent space to $\mathcal{M}_p$ at $Y \in \mathcal{M}_p$, denoted $\mathrm{T}_Y \mathcal{M}_p$, is the following subspace of $\mathbb{R}^{n \times p}$:*

$$\mathrm{T}_Y \mathcal{M}_p = \left\{ U \in \mathbb{R}^{n \times p} : \langle Y_i, U_i \rangle = 0 \, for \, i \in [n] \right\}.$$

Here, $Y_i \in \mathbb{R}^p$, $U_i \in \mathbb{R}^p$ *denote the $i$th rows of $Y$ and $U$ respectively, taken as column vectors. (In other words, $\mathrm{T}_Y \mathcal{M}_p$ is the space of matrices row-wise orthogonal to $Y$.) We call an element of $\mathrm{T}_Y \mathcal{M}_p$ a tangent vector (at $Y$).*

The Riemannian gradient at $Y \in \mathcal{M}_p$, $\mathrm{grad} \, \mathrm{OBJ}(Y)$, is the orthogonal projection of the classical Euclidean gradient at $Y$ onto $\mathrm{T}_Y \mathcal{M}_p$, which yields the following expression (see Appendix B):

$$\mathrm{grad} \, \mathrm{OBJ}(Y) = 2(A - \mathrm{diag}(\nu))Y, \quad \text{where } \nu_i := \sum_{j=1}^{n} A_{ij} \langle Y_i, Y_j \rangle, \text{ for all } i \in [n]. \quad (1)$$

(Note that $\nu \in \mathbb{R}^n$ is a function of $Y$, although we write $\nu$ instead of, e.g., $\nu(Y)$ when $Y$ is clear from context.) We take (1) to be the definition of $\nu$ from now on.

The Riemannian Hessian of OBJ at $Y \in \mathcal{M}_p$, $\mathrm{Hess} \, \mathrm{OBJ}(Y)$, is a linear, symmetric map from $\mathrm{T}_Y \mathcal{M}_p$ to $\mathrm{T}_Y \mathcal{M}_p$ given by the classical differential of (a smooth extension of) $\mathrm{grad} \, \mathrm{OBJ}(Y)$, projected to the tangent space [Bou22, Cor. 5.16]. This yields (see Appendix B):

$$\mathrm{Hess} \, \mathrm{OBJ}(Y)[U] = 2\mathrm{Proj}_Y \Big( (A - \mathrm{diag}(\nu))U \Big)$$

for $U \in \mathrm{T}_Y \mathcal{M}_p$, where the linear map $\mathrm{Proj}_Y : \mathbb{R}^{n \times p} \to \mathrm{T}_Y \mathcal{M}_p$ denotes the orthogonal projector onto $\mathrm{T}_Y \mathcal{M}_p \subseteq \mathbb{R}^{n \times p}$, i.e., $\mathrm{Proj}_Y(Z) = \mathrm{argmin}_{U \in \mathrm{T}_Y \mathcal{M}_p} \|U - Z\|$. In what follows, $\mathrm{Hess} \, \mathrm{OBJ}(Y)$ will only appear as part of a quadratic form, yielding the following cleaner expression:

$$\langle \mathrm{Hess} \, \mathrm{OBJ}(Y)[U], U \rangle = \langle 2\mathrm{Proj}_Y \left( (A - \mathrm{diag}(\nu))U \right), U \rangle = 2 \left\langle A - \mathrm{diag}(\nu), UU^\top \right\rangle,$$

for $U \in \mathrm{T}_Y \mathcal{M}_p$, where we used the fact that $\langle \mathrm{Proj}_Y(Z), U \rangle = \langle Z, U \rangle$ for any $Z \in \mathbb{R}^{n \times p}$, $U \in \mathrm{T}_Y \mathcal{M}_p$.

## 2.2 Necessary and sufficient conditions

Recall the definition of a local (and global) minimum:

**Definition 1** (Local/global minimum)**.** *Consider the program*

$$\min_{x \in D} f(x)$$

*where $D \subseteq \mathbb{R}^d$. $x \in D$ is a local minimum if there exists $\epsilon > 0$ such that if $y \in D$ and $\|y - x\| < \epsilon$, we have $f(x) \leq f(y)$. $x \in D$ is a global minimum if $f(x) \leq f(y)$ for all $y \in D$.*

The following are standard necessary conditions for local optimality, and correspond to the first and second-order critical point criteria that need to be satisfied by any local minimum of (MC-BM) (see [BVB18, Prop 2.4]).

**Proposition 2** (First-order critical point [BVB18, Def. 2.3] and [WW20, Prop. 3])**.** *$Y \in \mathcal{M}_p$ is a first-order critical point for (MC-BM) if and only if $\mathrm{grad} \, \mathrm{OBJ}(Y) = 2(A - \mathrm{diag}(\nu))Y = 0$. Equivalently, $Y \in \mathcal{M}_p$ is a first-order critical point if and only if there exists $\lambda \in \mathbb{R}^n$ such that $(A - \mathrm{diag}(\lambda))Y = 0$. If such a $\lambda$ exists, it is unique and equal to $\nu$ given by (1).*

**Proposition 3** (Second-order critical point [WW20, Prop. 4])**.** *A first-order critical point $Y \in \mathcal{M}_p$ is additionally a second-order critical point if and only if*

$$\langle \mathrm{Hess} \, \mathrm{OBJ}(Y)[U], U \rangle = 2 \left\langle A - \mathrm{diag}(\nu), UU^\top \right\rangle \geq 0$$

*for all $U \in \mathrm{T}_Y \mathcal{M}_p$, and where $\nu$ is given in (1). (This is equivalent to $\mathrm{Hess} \, \mathrm{OBJ}(Y) \succeq 0$.)*

We say a critical point or local minimum is *spurious* if it is not globally optimal. Finally, we characterize which first-order critical points of (MC-BM) are globally optimal. Since second-order critical points and local minima are also first-order critical points, this also provides a characterization of optimality for them.

**Proposition 4** (Characterization of optimality for first-order critical points)**.** *A first-order critical point $Y$ of (MC-BM) is globally optimal if and only if $A - \mathrm{diag}(\nu) \succeq 0$, where $\nu$ is given in (1).*

While not framed precisely in this way, Proposition 4 follows directly from prior work (see Appendix B for details). The proof involves a comparison between the criticality conditions of (MC-BM) and the Karush–Kuhn–Tucker (KKT) conditions of (MC-SDP).

## 2.3 Riemannian gradient descent

The analogue to gradient descent for optimizing over a smooth manifold is Riemannian gradient descent (see [Bou22, Ch. 4] for an introduction), which yields analogous analyses and guarantees. It includes as part of its specification a *retraction* [Bou22, Def. 3.47]. A rectraction on $\mathcal{M}_p$ associates to each point $Y \in \mathcal{M}_p$ a map $\mathrm{R}_Y : \mathrm{T}_Y\mathcal{M}_p \to \mathcal{M}_p$ which converts movement in the tangent space to movement on the manifold $\mathcal{M}_p$. We use the natural *metric projection retraction* [Bou22, Sec. 5.12], defined for $Y \in \mathcal{M}_p, U \in \mathrm{T}_Y\mathcal{M}_p$ as $\mathrm{R}_Y(U) := \mathrm{argmin}_{Z \in \mathcal{M}_p} \|(Y + U) - Z\|$. With this definition, it is easy to see that $\mathrm{R}_Y(U)$ is $Y + U$ followed by a normalization of each row. This yields the following Riemannian gradient descent algorithm for (MC-BM):

**Input:** Initializer $Y^{(0)} \in \mathcal{M}_p$, step size $\eta > 0$.

**For** $t = 0, 1, 2, \ldots$

$$Y^{(t+1)} = \mathrm{R}_{Y^{(t)}} \left( -\eta \mathrm{grad}\, \mathrm{OBJ}(Y^{(t)}) \right).$$

## 2.4 Construction-specific definitions

Finally, we give a few miscellaneous technical definitions which will be used in our construction of a spurious local minimum for (MC-BM).

**Definition 2** (Axial position). *We call the matrix*

$$\widetilde{Y} := \begin{bmatrix} I_{(n/2)} \\ -I_{(n/2)} \end{bmatrix} \in \mathcal{M}_{n,(n/2)}$$

*the axial position, where $I_{n/2}$ denotes the identity matrix in $\mathbb{R}^{\frac{n}{2} \times \frac{n}{2}}$.*

We use the term "axial position" because when we view $\mathcal{M}_{n,(n/2)}$ as a Cartesian product of $n$ unit spheres in $\mathbb{R}^{n/2}$, $\widetilde{Y}$ corresponds to placing a single unit vector on both the negative and positive sides of each axis in $\mathbb{R}^{n/2}$.

The following sets of matrices will be important in our construction:

**Definition 3** (Pseudo-PD, pseudo-PSD). *We say a matrix $M \in \mathbb{S}^{n \times n}$ is pseudo-PD ("pseudo-positive definite") if $M[i] \succ 0$ for all $i \in [n]$, where $M[i] \in \mathbb{S}^{(n-1) \times (n-1)}$ denotes the submatrix of $M$ formed by removing the $i$th row and column. Similarly, we say that $M \in \mathbb{S}^{n \times n}$ is pseudo-PSD ("pseudo-positive semidefinite") if $M[i] \succeq 0$ for all $i \in [n]$.*

**Definition 4** (Strictly pseudo-PD, strictly pseudo-PSD). *We say a matrix is $M \in \mathbb{S}^{n \times n}$ is strictly pseudo-PD if it is pseudo-PD but not positive semidefinite. We say $M$ is strictly pseudo-PSD if it is pseudo-PSD but not positive semidefinite. (Note that in both cases we require $M \not\succeq 0$, not $M \not\succ 0$.)*

Clearly every strictly pseudo-PD matrix is also strictly pseudo-PSD, but the converse turns out to be false.

## 3 Proof of Theorem 1

In this section we prove Theorem 1. We focus on the case where $n$ is even and $p = n/2$ and construct cost matrices for which $\widetilde{Y}$ is a spurious local minimum, since constructions for $p < n/2$ can be easily extracted from the former by padding with zeros. Before this, as a warm-up, we characterize those cost matrices for which $\widetilde{Y}$ is a spurious first and second-order critical point in the following two propositions. While not strictly necessary for the proof of Theorem 1, Propositions 5 and 6 have far simpler proofs and are interesting in their own right. An overview of the proofs is given in Section 4, and the full proofs can be found in Appendix C.

**Proposition 5** (First-order critical point characterization for $\widetilde{Y}$). *For (MC-BM) when $p = n/2$, the axial position $\widetilde{Y}$ is a first-order critical point if and only if the cost matrix $A$ takes the form*

$$A = \begin{bmatrix} B & B \\ B & B \end{bmatrix} + \mathrm{diag}(\alpha) \tag{2}$$

*for some $\alpha \in \mathbb{R}^n$ and $B \in \mathbb{S}^{\frac{n}{2} \times \frac{n}{2}}$. Furthermore, $\widetilde{Y}$ is additionally spurious if and only if $B \not\succeq 0$.*

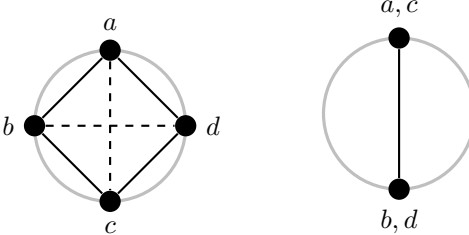

Figure 2: A spurious minimum and corresponding global minimum for $n = 4, p = 2$.

**Proposition 6** (Second-order critical point characterization for $\widetilde{Y}$). *For* (MC-BM) *when* $p = n/2$, *the axial position $\widetilde{Y}$ is a second-order critical point if and only if the cost matrix $A$ takes the form*

$$A = \begin{bmatrix} P & P \\ P & P \end{bmatrix} + \operatorname{diag}(\alpha)$$

*for some $\alpha \in \mathbb{R}^n$ and pseudo-PSD $P \in \mathbb{S}^{\frac{n}{2} \times \frac{n}{2}}$. Furthermore, $\widetilde{Y}$ is additionally spurious if and only if $P$ is strictly pseudo-PSD.*

It is not surprising that Propositions 5 and 6 allow you to arbitrarily change the diagonal of the cost matrix. Doing so simply corresponds to adding a constant to the objective value at each point and does not change the geometry of the problem.[3]

Next, the following lemma provides a sufficient condition for $\widetilde{Y}$ to be a spurious local minimum. This is the most challenging of our results to prove, and we give the proof and discuss the challenges involved in Section 5. Key sublemmas are left to Appendix D.

**Lemma 1** (Local minimum condition for $\widetilde{Y}$). *For* (MC-BM) *when* $p = n/2$, *the axial position $\widetilde{Y}$ is a local minimum if the cost matrix $A$ takes the form*

$$A = \begin{bmatrix} M & M \\ M & M \end{bmatrix} + \operatorname{diag}(\alpha) \tag{3}$$

*for some $\alpha \in \mathbb{R}^n$ and pseudo-PD $M \in \mathbb{S}^{\frac{n}{2} \times \frac{n}{2}}$. Furthermore, $\widetilde{Y}$ is additionally spurious if $M$ is strictly pseudo-PD.*

Actualizing Lemma 1 to construct a spurious local minimum requires the existence of strictly pseudo-PD matrices, which we posit in the following lemma:

**Lemma 2** (Existence of strictly pseudo-PD matrices). *The set of $k \times k$ strictly pseudo-PD matrices is nonempty for any $k \geq 2$.*

We provide constructions of strictly pseudo-PD matrices in Appendix E. Our main nonnegative construction takes the form $UU^\top - \epsilon I_k$, where $U \in \mathbb{R}^{k \times (k-1)}$ is a random matrix. It is shown $(UU^\top)[i] \succ 0$ for all $i$ with high probability. One can show that when $\epsilon > 0$ is sufficiently small, the $(k-1) \times (k-1)$ principal submatrices of $UU^\top - \epsilon I_k$ remain positive definite, while $UU^\top - \epsilon I_k \nsucceq 0$ since $UU^\top$ is rank-deficient.

We use Lemmas 1 and 2 to prove Theorem 1:

*Proof of Theorem 1.* Lemmas 1 and 2 imply that for even $n$ such that $n \geq 4$, there exists an instance of (MC-BM) with a spurious local minimum when $p = n/2$. To construct a spurious local minimum for $p < n/2$, we can simply use the construction for $n' = 2p$ vertices and rank $p$. The entries of the cost matrix that correspond to additional rows of elements of $\mathcal{M}_{n,p}$ past $n'$ can be set to 0, ensuring that these additional rows cannot affect the objective value. (See Appendix F for details.) $\square$

One instantiation of our construction for $n = 4, p = 2$ is illustrated in Figure 2. In this visualization, we think of the nonnegative cost matrix as the adjacency matrix of a weighted graph, which is natural

---

[3]One can easily check that the Riemannian derivatives at any point $Y \in \mathcal{M}_p$ remain unchanged since $\nu$ in (1) will act as an offset.

for (MC-BM). Each row of $Y$ specifies the position of one of the vertices on the unit sphere in $\mathbb{R}^p$ (for $p = 2$, the circle shown in gray). We illustrate "heavy" (higher weight) edges with solid lines, and "light" (lower weight) edges with dashed lines. Intuitively, each edge "pushes" its endpoints away from each other, with heavier edges pushing harder. We see by symmetry that this state is in equilibrium (each vertex is pushed equally clockwise and counter-clockwise), so the gradient is 0. Showing that this instance is indeed a local minimum is much more involved, and requires arguing that if the vertices were perturbed slightly, the heavy edges would still approximately cancel out, and the main force on the vertices would be the light edges pushing the pairs $a, c$ and $b, d$ back to being diametrically opposite.[4]

## 4    Overview of the proofs of criticality (Propositions 5 and 6)

In this section we provide overviews of the proofs of Propositions 5 and 6, which characterize the cost matrices for which $\widetilde{Y}$ is a spurious/optimal first and second-order critical point respectively (see Appendix C for the full proofs).

### 4.1    Overview of the proof of first-order criticality (Proposition 5)

Supposing the cost matrix $A$ takes the form (2), one can set $\lambda \leftarrow \alpha$ where $\lambda \in \mathbb{R}^n$ is the multiplier from Proposition 2, and then observe $(A - \mathrm{diag}(\lambda))\widetilde{Y} = 0$, implying $\widetilde{Y}$ is a first-order critical point. For the other direction, one can show that if $\widetilde{Y}$ is a first-order critical point with associated multiplier $\lambda \in \mathbb{R}^n$ (from Proposition 2), then $(A - \mathrm{diag}(\lambda))\widetilde{Y} = 0$ implies $A - \mathrm{diag}(\lambda)$ must take the block form $\begin{bmatrix} B & B \\ B & B \end{bmatrix}$ for some $B \in \mathbb{S}^{\frac{n}{2} \times \frac{n}{2}}$. Thus, $A$ takes the form (2) with $\alpha \leftarrow \lambda$.

To show $\widetilde{Y}$ is additionally spurious if and only if $B \not\succeq 0$, note from above that if $Y$ is a first-order critical point, the unique associated multiplier $\lambda$ is equal to $\alpha$. Recall from Proposition 2 that $\nu = \lambda$ at a first-order critical point, implying $A - \mathrm{diag}(\nu) = A - \mathrm{diag}(\alpha) = \begin{bmatrix} B & B \\ B & B \end{bmatrix}$. Such a matrix is positive semidefinite if and only if $B$ is positive semidefinite, and Proposition 4 concludes the proof.

### 4.2    Overview of the proof of second-order criticality (Proposition 6)

Since every second-order critical point is a first-order critical point, we assume the cost matrix $A$ takes the form (2) and show that $\widetilde{Y}$ is additionally a second-order critical point if and only if $B$ is additionally pseudo-PSD. Recall from Section 4.1 that when $\widetilde{Y}$ is a first-order critical point, $A - \mathrm{diag}(\nu) = \begin{bmatrix} B & B \\ B & B \end{bmatrix}$. Now, one can observe that an arbitrary element $U \in \mathrm{T}_{\widetilde{Y}}\mathcal{M}_{n/2}$ must take the following form: $U_{ii} = U_{(n/2)+i,i} = 0$ for all $i \in [n/2]$, and all other entries of $U$ are arbitrary. Due to this, one can show that

$$\left\langle \mathrm{Hess}\,\mathrm{OBJ}(\widetilde{Y})[U], U \right\rangle = 2 \left\langle A - \mathrm{diag}(\nu), UU^\top \right\rangle \geq 0 \quad \text{for all } U \in \mathrm{T}_{\widetilde{Y}}\mathcal{M}_{n/2}$$

$$\iff 2 \left\langle \begin{bmatrix} B & B \\ B & B \end{bmatrix}, uu^\top \right\rangle \geq 0 \quad \text{for all } u \in R_1 \cup \cdots \cup R_{n/2}, \tag{4}$$

where $R_i$ is the subspace of $\mathbb{R}^n$ consisting of those vectors $v$ such that $v_i = v_{(n/2)+i} = 0$. Restricting $u$ to only lie in $R_i$ in (4) is equivalent to the matrix $\begin{bmatrix} B[i] & B[i] \\ B[i] & B[i] \end{bmatrix} \in \mathbb{S}^{(n-2)\times(n-2)}$ being positive semidefinite, which is equivalent to $B[i]$ being positive semidefinite. (Recall that $B[i]$ denotes the submatrix of $B$ formed by removing row $i$ and column $i$.) Then (4), which takes the preceding statement over all $i \in [n/2]$, is equivalent to $B$ being pseudo-PSD. (We note that the preceding portion of the proof is formalized in a slightly different way in Appendix C.) The second half of Proposition 6 (characterizing when the second-order critical point $\widetilde{Y}$ is spurious) follows immediately

---

[4]We provide an interactive visualization for the $p = 2$ and $p = 3$ cases at
`https://vaidehi8913.github.io/burer-monteiro`.

from the corresponding characterization in Proposition 5, since every second-order critical point is a first-order critical point.

# 5 Proof of local minimality (Lemma 1)

In this section we give the proof of Lemma 1. We first discuss the challenges involved and why classical techniques break down. We then state the two main sublemmas used in the proof of Lemma 1 (with proofs of these sublemmas left to Appendix D). Finally, we give the proof of Lemma 1 itself.

**Challenges.** Unfortunately, arguing about the value of the objective function at some point $Y$ near $\widetilde{Y}$ is challenging, and classical techniques for proving that $\widetilde{Y}$ is a local minimum fail. For example, [WW20], which constructs spurious local minima for (MC-BM) when $p < \sqrt{2n}$, similarly first constructs spurious second-order critical points and then proves that they are additionally local minima. However, their proof follows because their spurious second-order critical points are non-degenerate [WW20, Def. 3], which corresponds to the rank of the Riemannian Hessian being sufficiently high. We show in Appendix D (Proposition 9) that for *any* instance of (MC-BM) when $p \geq \sqrt{2n}$, *all* spurious second-order critical points are degenerate, meaning this approach will not work. Furthermore, there is no hope of using the positivity of a higher-order Riemannian derivative (e.g., the fourth derivative) to prove that $\widetilde{Y}$ is a local minimum, since it can be shown that all higher-order derivatives are degenerate. (See Appendix D for further discussion of these challenges.)

**Overview and key sublemmas.** Thus, we provide a novel approach involving Riemannian gradient descent (Section 2.3). For the first sublemma below, recall that a neighborhood of a point $Y \in \mathcal{M}_p$ is a set of the form $\{Y' \in \mathcal{M}_p : ||Y - Y'|| < \epsilon\}$ for some $\epsilon > 0$. The proof is given in Appendix D.

**Lemma 3** (Convergence to a point with the same objective value). *In the setting of Lemma 1 with pseudo-PD $M \in \mathbb{S}^{\frac{n}{2} \times \frac{n}{2}}$, there exists a neighborhood $N \subseteq \mathcal{M}_{n/2}$ of $\widetilde{Y}$ and $\eta' > 0$ depending only on the instance of* (MC-BM) *such that if you initialize Riemannian gradient descent (as specified in Section 2.3) with any step size $\eta < \eta'$ at any point $Y^{(0)} \in N$, it converges to a point $\overline{Y}$ such that* $\text{OBJ}(\overline{Y}) = \text{OBJ}(\widetilde{Y})$.

Note that Lemma 3 does not imply convergence to $\widetilde{Y}$ itself; this is not actually true due to the degeneracy mentioned above. Instead, we show in the proof of Lemma 3 that $\overline{Y}$ is an *antipodal configuration*, i.e., it takes the form $\begin{bmatrix} G \\ -G \end{bmatrix} \in \mathcal{M}_{n/2}$ for some $G \in \mathbb{R}^{\frac{n}{2} \times \frac{n}{2}}$. Such antipodal configurations (of which $\widetilde{Y}$ is one) all have the same objective value and correspond in particular to "flat" directions from $\widetilde{Y}$. For a given $Y^{(0)}$, it is not a priori clear which of these antipodal configurations it will converge to, so we argue convergence to *some* antipodal configuration.

In the proof of Lemma 3, we track convergence to an antipodal configuration via a potential $\Phi : \mathcal{M}_{n/2} \to \mathbb{R}_{\geq 0}$, where $\Phi\left(\begin{bmatrix} G_1 \\ G_2 \end{bmatrix}\right) := ||G_1 + G_2||^2$. (Here, $G_1, G_2 \in \mathbb{R}^{\frac{n}{2} \times \frac{n}{2}}$.) Clearly $\Phi$ is $0$ if and only if the input is antipodal. We show that $\Phi$ decreases geometrically over the iterations of Riemannian gradient descent.

Next, we show via a smoothness argument that with sufficiently small step size, the objective is nonincreasing over the iterations of Riemannian gradient descent. The proof is given in Appendix D.

**Lemma 4** (OBJ is nonincreasing). *There exists $\tilde{\eta} > 0$ depending only on the instance of* (MC-BM) *such that a single iteration of Riemannian gradient descent (as specified in Section 2.3) with any step size $\eta < \tilde{\eta}$ cannot increase the objective value, regardless of the starting point.*

Lemmas 3 and 4 imply $\text{OBJ}(Y^{(0)}) \leq \text{OBJ}(\overline{Y}) = \text{OBJ}(\widetilde{Y})$, and as a result the neighborhood $N$ in Lemma 3 certifies that $\widetilde{Y}$ is a local minimum. The details follow:

*Proof of Lemma 1.* We first show that if $M \in \mathbb{S}^{\frac{n}{2} \times \frac{n}{2}}$ is pseudo-PD, then $\widetilde{Y}$ is a local minimum. We claim that the neighborhood $N$ from Lemma 3 certifies that $\widetilde{Y}$ is a local minimum. Indeed, let $V \in N$, and initialize Riemannian gradient descent at $V$ with step size $\eta < \min\{\eta', \tilde{\eta}\}$, with $\eta', \tilde{\eta}$ defined as in Lemmas 3, 4. Per Lemma 3, we know that Riemannian gradient descent will converge to

a point $\overline{V}$ such that $\text{OBJ}(\widetilde{Y}) = \text{OBJ}(\overline{V})$. Since $\text{OBJ}$ is continuous, convergence in iterates translates to convergence in objective values, and thus the nonincreasing nature of Riemannian gradient descent from Lemma 4 implies $\text{OBJ}(\overline{V}) \leq \text{OBJ}(V)$. Then $\text{OBJ}(\widetilde{Y}) = \text{OBJ}(\overline{V}) \leq \text{OBJ}(V)$, and we are done.

Now we show that $\widetilde{Y}$ is additionally spurious if $M$ is strictly pseudo-PD. Indeed, this follows immediately from the last line of Proposition 5. (Recall that any local minimum is also a first-order critical point.) $\qquad\square$

## 6 Conclusion

We show that the Burer-Monteiro method can fail for instances of the Max-Cut SDP, for rank up to $p = \frac{n}{2}$. To the best of our knowledge, prior to our work it was unknown whether the Burer-Monteiro method could fail for any instance of any SDP with rank above the Barvinok-Pataki bound. We settle this question and thus justify the use of smoothed analysis to obtain guarantees for the Burer-Monteiro method.

There are many interesting potential follow-up directions to this work. We provide one construction of (MC-BM) instances with spurious local minima. Our construction has, and relies on, many interesting properties and symmetries. It is possible that some of these properties are necessary, and further analyzing this construction could give insight that allows us to fully characterize spurious minima for (MC-BM) instances. Analyzing this construction could also help us understand this interesting threshold phenomenon for (MC-BM) when $p = n/2$—one dimension higher and there are not only no spurious local minima, there are no spurious second-order critical points at all. Another potential direction is seeing if similar techniques can be used to construct instances with spurious local minima for SDPs with other structures (not just Max-Cut).

Lastly, we note that a limitation of our work is that it only points to the existence of local minima, and does not give a full characterization of when we can expect local minima to exist. We also note that since this is a theoretical result about optimization landscapes, we do not foresee any adverse societal impacts of our work.

## Acknowledgments and Disclosure of Funding

LO was supported by a National Science Foundation (NSF) REU supplement associated with CCF-1934931 and an undergraduate research grant from Northwestern University. VS and AV were also supported by the NSF under Grant Nos. CCF-1652491, CCF-1934931, and EECS-2216970. The authors thank the NeurIPS 2022 reviewers for helpful suggestions.

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
