# Appendix

## Table of Contents

## A  Further discussion of prior work

The two most relevant papers to this work are [BVB18] and [WW20]. [BVB18] excludes the presence of spurious second-order critical points for (MC-BM) outside of a measure-zero set of cost matrices when $\frac{p(p+1)}{2} > n$. (Furthermore, their result extends to a broad class of smooth SDPs.) Additionally, [BVB18] shows that when $p > n/2$, (MC-BM) has no spurious second-order critical points. [WW20] tightens the main lower bound of [BVB18] to $\frac{p(p+1)}{2} + p > n$ and also shows that when $\frac{p(p+1)}{2} + p \leq n$, there exists a set of cost matrices with non-zero measure whose corresponding instances of (MC-BM) have spurious local minima.

It was open to the best of our knowledge whether there exists any instance of (MC-BM) with spurious second-order critical points when $\frac{p(p+1)}{2} + p > n$. (In fact, this question was open for the broad class of smooth SDPs analyzed in [BVB18]—see Section 6 in that paper.) We note that it is not clear how to extend the techniques of [WW20] to the setting of our paper since their constructions critically rely on a technical assumption (the existence of "minimally secant" matrices) which provably never holds when $\frac{p(p+1)}{2} > n$ (as they note in Appendix B). As a result, our paper takes a different approach.

There has also been a line of work [BBJN18, PJB18, CM19] seeking to provide polynomial-time convergence guarantees to approximate global optima in a smoothed analysis setting. [BBJN18] in particular performs smoothed analysis on an unconstrained quadratically-penalized version of (MC-BM) (and its generalizations) and also provides a lower bound. However, their lower-bound construction does not apply to (MC-BM) itself (or its generalizations). (In particular, their lower-bound construction sets the cost matrix to be 0, which does not work in our setting.)

Finally, [MMMO17, Thm. 1] implies that when the cost matrix $A$ comes from a weighted graph with nonnegative weights on the edges (as is typical in Max-Cut relaxations), any spurious local

minimum still achieves the globally optimal value up to an $O(1/p)$ multiplicative error. We note that as mentioned in Section 1, there are many applications of (MC-SDP) where $A$ may have negative entries and where the solution value is not as important as the optimal SDP solution itself (which is used to recover some ground-truth solution). As we showed in Proposition 13, there exist instances of (MC-SDP) where the optimal solution and the spurious point $\widetilde{Y}\widetilde{Y}^\top$ are qualitatively very different.

## B Toolbox

### B.1 Proof of Proposition 4

Proposition 4 follows directly from Corollary 2.9 and Proposition 2.10 from [BVB18], which give analogous claims for a more general class of programs. In regards to Corollary 2.9 (the "if" direction), note that (MC-BM) trivially satisfies Assumption 1.1a (which is equivalent to the linear independence constraint qualification, aka LICQ, holding over the entire feasible region). Proposition 2.10 (the "only if" direction) requires strong duality, and strong duality holds for the convex program (MC-SDP) since it satisfies Slater's condition.

### B.2 Riemannian gradient and Riemannian Hessian for (MC-BM)

**Proposition 7** (Riemannian gradient for (MC-BM))**.** *The Riemannian gradient of* OBJ *at* $Y \in \mathcal{M}_p$ *is given by*

$$\operatorname{grad} \operatorname{OBJ}(Y) = 2(A - \operatorname{diag}(\nu))Y, \quad \text{where } \nu_i := \sum_{j=1}^{n} A_{ij} \langle Y_i, Y_j \rangle, \text{ for all } i \in [n]. \qquad (5)$$

*Here,* $Y_i \in \mathbb{R}^p$ *denotes the ith row of* $Y$*, taken as a column vector.*

*Proof.* For a smooth objective function over a Riemannian submanifold of a vector space [Bou22, Def. 3.55], the Riemannian gradient is given by the orthogonal projection of the Euclidean gradient to the tangent space [Bou22, Prop. 3.61]. Since $\mathcal{M}_p$ is a Riemannian submanifold of $\mathbb{R}^{n \times p}$ [BVB18, Sec. 2.1], applying this yields

$$\operatorname{grad} \operatorname{OBJ}(Y) = \operatorname{Proj}_Y(2AY) = 2\operatorname{Proj}_Y(AY), \qquad (6)$$

where the linear map $\operatorname{Proj}_Y : \mathbb{R}^{n \times p} \to \mathrm{T}_Y \mathcal{M}_p$ denotes the orthogonal projector onto $\mathrm{T}_Y \mathcal{M}_p \subseteq \mathbb{R}^{n \times p}$, i.e., $\operatorname{Proj}_Y(Z) = \operatorname{argmin}_{U \in \mathrm{T}_Y \mathcal{M}_p} \|U - Z\|$. Since $\mathrm{T}_Y \mathcal{M}_p$ consists of those matrices in $\mathbb{R}^{n \times p}$ which are row-wise orthogonal to $Y$ (Proposition 1), it is clear that the orthogonal projection of $Z \in \mathbb{R}^{n \times p}$ onto $\mathrm{T}_Y \mathcal{M}_p$ is found by going row by row over $Z$ and each time deleting the component of row $i$ of $Z$ which lies in the span of row $i$ of $Y$. In other words,

$$\operatorname{Proj}_Y(Z) = Z - \operatorname{diag}(\mu)Y, \quad \text{where } \mu_i := \langle Z_i, Y_i \rangle, \text{ for all } i \in [n]. \qquad (7)$$

(6) and (7) together yield our result. $\qquad \square$

The Riemannian Hessian of OBJ at $Y \in \mathcal{M}_p$, $\operatorname{Hess} \operatorname{OBJ}(Y)$, is a linear, symmetric map from $\mathrm{T}_Y \mathcal{M}_p$ to $\mathrm{T}_Y \mathcal{M}_p$. For a Riemannian submanifold of a vector space such as $\mathcal{M}_p$, this is given by the classical differential of (a smooth extension of) $\operatorname{grad} \operatorname{OBJ}(Y)$, projected to the tangent space [Bou22, Cor. 5.16].

We note that the Riemannian Hessian is the natural Riemannian analog of the Euclidean Hessian, and while the Euclidean Hessian $\nabla^2 f(x)$ of a function $f : \mathbb{R}^m \to \mathbb{R}$ at $x \in \mathbb{R}^m$ can be thought of as a symmetric $m \times m$ matrix containing the second-order partial derivatives of $f$ at $x$, it is often best understood as a linear map $\nabla^2 f(x) : \mathbb{R}^m \to \mathbb{R}^m$ defined via $\nabla^2 f(x)[u] = Hu$, where $H \in \mathbb{S}^{m \times m}$ is the aforementioned "matrix form" of $\nabla^2 f(x)$. With this viewpoint, $\nabla^2 f(x)[u]$ is the directional derivative of the Euclidean gradient $\nabla f(x)$ in the direction $u$. Similarly, while the Riemannian Hessian of OBJ at $Y \in \mathcal{M}_p$, $\operatorname{Hess} \operatorname{OBJ}(Y)$, could be identified with a symmetric $\dim(\mathcal{M}_p) \times \dim(\mathcal{M}_p)$ matrix,[5] it is best understood as a linear map $\operatorname{Hess} \operatorname{OBJ}(Y) : \mathrm{T}_Y \mathcal{M}_p \to$

---

[5]Note that for a smooth manifold $\mathcal{M}$, $\dim(\mathcal{M})$ is defined as $\dim(\mathrm{T}_x \mathcal{M})$, where $\mathrm{T}_x \mathcal{M}$ denotes the tangent space (a vector space) at $x \in \mathcal{M}$. ($\dim(\mathrm{T}_x \mathcal{M})$ is independent of $x$.)

$\mathrm{T}_Y \mathcal{M}_p$, where $\mathrm{Hess\,OBJ}(Y)[U]$ denotes the "directional derivative" of the Riemannian gradient in the direction $U$. (The correct definition of "directional derivative" in this case is the Riemannian connection [Bou22, Thm. 5.6].) As is standard, we take the latter form to be the definition of the Riemannian Hessian [Bou22, Def. 5.14].

**Proposition 8** (Riemannian Hessian for (MC-BM)). *The Riemannian Hessian of* OBJ *at* $Y \in \mathcal{M}_p$, *acting on* $U \in \mathrm{T}_Y \mathcal{M}_p$, *is given by*

$$\mathrm{Hess\,OBJ}(Y)[U] = 2\mathrm{Proj}_Y \left( (A - \mathrm{diag}(\nu))U \right),$$

*where the linear map* $\mathrm{Proj}_Y : \mathbb{R}^{n \times p} \to \mathrm{T}_Y \mathcal{M}_p$ *denotes the orthogonal projector onto* $\mathrm{T}_Y \mathcal{M}_p \subseteq \mathbb{R}^{n \times p}$, *i.e.,* $\mathrm{Proj}_Y(Z) = \mathrm{argmin}_{U \in \mathrm{T}_Y \mathcal{M}_p} \|U - Z\|$. $\nu$ *is defined as in* (5).

*Proof.* This follows immediately from Equation 2.7 in [BVB18], which provides an expression for the Riemannian Hessian for a more general class of programs. (Their expression for the multiplier $\nu$, which they call $\mu$—see Equation 2.5 in that paper, is more complicated as it is for a more general class of programs, which is why we derived the Riemannian gradient from first principles in the proof of Proposition 7. However, the two expressions for this multiplier are ultimately equivalent for (MC-BM) due to the uniqueness of the Riemannian gradient.) See also Section 7.7 of [Bou22] for exposition (including an expression for the Riemannian Hessian) for a very general class of programs encompassing (MC-BM). $\square$

## C Proofs of criticality (Propositions 5 and 6)

### C.1 Proof of Proposition 5

We first prove the first half of Proposition 5, which characterizes when $\widetilde{Y}$ is a first-order critical point. If $A$ takes the form (2) for some $\alpha \in \mathbb{R}^n$ and $B \in \mathbb{S}^{\frac{n}{2} \times \frac{n}{2}}$, we can set $\lambda \leftarrow \alpha$ where $\lambda$ is our choice for the multiplier from Proposition 2. Observe then that $(A - \mathrm{diag}(\lambda))\widetilde{Y} = 0$, implying $\widetilde{Y}$ is indeed a first-order critical point. For the other direction, suppose that $\widetilde{Y}$ is a first-order critical point with associated multiplier $\lambda$ (from Proposition 2), and consider the matrix $S := A - \mathrm{diag}(\lambda) \in \mathbb{S}^{n \times n}$, which can be expressed in the block form

$$S = \begin{bmatrix} S_1 & S_2 \\ S_2^\top & S_3 \end{bmatrix}$$

for some $S_1, S_3 \in \mathbb{S}^{\frac{n}{2} \times \frac{n}{2}}$ and $S_2 \in \mathbb{R}^{\frac{n}{2} \times \frac{n}{2}}$. Then $S\widetilde{Y} = 0$ implies $S_1 - S_2 = 0$ and $S_2^\top - S_3 = 0$, and thus $S_1 = S_2 = S_3$. Thus, $A = S + \mathrm{diag}(\lambda)$ indeed takes the form (2).

Now we prove the second half of Proposition 5: the characterization of when $\widetilde{Y}$ is a spurious first-order critical point. Supposing that the cost matrix takes the form (2) for some $\alpha \in \mathbb{R}^n$ and $B \in \mathbb{S}^{\frac{n}{2} \times \frac{n}{2}}$ (as we've shown is necessary for $\widetilde{Y}$ to be a first-order critical point), we show that $\widetilde{Y}$ is additionally spurious if and only if $B \not\succeq 0$. Recall from above that the unique multiplier $\lambda \in \mathbb{R}^n$ (from Proposition 2) associated with $\widetilde{Y}$ is precisely $\alpha$. Then it follows from Proposition 4 that $\widetilde{Y}$ is spurious if and only if

$$A - \mathrm{diag}(\lambda) = \begin{bmatrix} B & B \\ B & B \end{bmatrix}$$

is not positive semidefinite. (Recall from Proposition 2 that $\nu = \lambda$ at a first-order critical point.) We claim

$$\begin{bmatrix} B & B \\ B & B \end{bmatrix} = \begin{bmatrix} 1 & 1 \\ 1 & 1 \end{bmatrix} \otimes B \not\succeq 0 \quad \Longleftrightarrow \quad B \not\succeq 0, \tag{8}$$

where $\otimes$ denotes the Kronecker product. Indeed, this follows because the spectrum of $F \otimes G$, denoted $\sigma(F \otimes G)$, for two square, real matrices $F, G$ is given by

$$\sigma(F \otimes G) = \{\lambda\mu : \lambda \in \sigma(F), \mu \in \sigma(G)\}.$$

This, combined with the fact that $\begin{bmatrix} 1 & 1 \\ 1 & 1 \end{bmatrix} \succeq 0$, implies (8).

## C.2  Proof of Proposition 6

We first prove the first half of Proposition 6, which characterizes when $\widetilde{Y}$ is a second-order critical point. Since any second-order critical point is also a first-order critical point, the first half of Proposition 5 implies it is necessary for the cost matrix $A$ to take the form (2) for some $\alpha \in \mathbb{R}^n$ and $B \in \mathbb{S}^{\frac{n}{2} \times \frac{n}{2}}$ for $\widetilde{Y}$ to be a second-order critical point. We will show that $\widetilde{Y}$ is additionally a second-order critical point if and only if the matrix $B$ from (2) is pseudo-PSD.

To this end, recall from the proof of Proposition 5 that the unique multiplier $\lambda$ associated with $\widetilde{Y}$ when $A$ takes the form (2) is precisely $\alpha$. Then, writing

$$S := A - \operatorname{diag}(\lambda) = \begin{bmatrix} B & B \\ B & B \end{bmatrix}, \tag{9}$$

clearly the condition for the second-order criticality of $\widetilde{Y}$ (Proposition 3) is equivalent to

$$\left\langle S, UU^\top \right\rangle \geq 0 \quad \text{for } U \in \mathrm{T}_{\widetilde{Y}} \mathcal{M}_{n/2}. \tag{10}$$

Define

$$O_{\text{off}} := \left\{ G \in \mathbb{R}^{\frac{n}{2} \times \frac{n}{2}} : \operatorname{diag}(G) = 0 \right\}$$

to be the subspace of $\mathbb{R}^{\frac{n}{2} \times \frac{n}{2}}$ consisting of matrices with zeros on their diagonals. Note then that $\mathrm{T}_{\widetilde{Y}} \mathcal{M}_{n/2}$ is precisely

$$\mathrm{T}_{\widetilde{Y}} \mathcal{M}_{n/2} = \left\{ \begin{bmatrix} U_1 \\ U_2 \end{bmatrix} : U_1, U_2 \in O_{\text{off}} \right\}. \tag{11}$$

In other words, $\mathrm{T}_{\widetilde{Y}} \mathcal{M}_{n/2}$ is the set of all matrices $U \in \mathbb{R}^{n \times \frac{n}{2}}$ such that $U_{ii} = U_{(n/2+i),i} = 0$ for all $i \in [n/2]$, and the other entries are completely arbitrary. Observe that (9) and (11) imply the second-order criticality condition (10) is equivalent to

$$\left\langle B, (U_1 + U_2)(U_1 + U_2)^\top \right\rangle \geq 0 \quad \text{for } U_1, U_2 \in O_{\text{off}}. \tag{12}$$

Next, note that (12) (and therefore (10)) is equivalent to

$$\left\langle B, GG^\top \right\rangle \geq 0 \quad \text{for } G \in O_{\text{off}} \tag{13}$$

since $O_{\text{off}}$ is closed under addition.

Thus, we have shown at this point that $\widetilde{Y}$ is a second-order critical point if and only if (13) holds. Now let $T_i$ for $i \in [n/2]$ denote the $(n/2 - 1)$-dimensional subspace of $\mathbb{R}^{n/2}$ obtained by fixing the $i$th entry to be 0 and letting all other entries vary arbitrarily. Observe that

$$\left\{ GG^\top : G \in O_{\text{off}} \right\} = \left\{ \sum_{k=1}^{n/2} v_k v_k^\top : v_i \in T_i \text{ for all } i \in [n/2] \right\}.$$

Thus, we can reexpress the second-order criticality condition (13) as follows:

$$\left\langle B, GG^\top \right\rangle \geq 0 \quad \text{for } G \in O_{\text{off}}$$
$$\iff \left\langle B, \sum_{k=1}^{n/2} v_k v_k^\top \right\rangle \geq 0 \quad \text{for } v_1, \in T_1, \dots, v_{n/2} \in T_{n/2}$$
$$\iff \left\langle B, vv^\top \right\rangle \geq 0 \quad \text{for } v \in T_1 \cup \cdots \cup T_{n/2}. \tag{14}$$

(14) is equivalent to $B$ being pseudo-PSD.

As for the second half of Proposition 6, the characterization of when $\widetilde{Y}$ is a spurious second-order critical point, this follows immediately from the first half of Proposition 6 and the characterization of when $\widetilde{Y}$ is a spurious first-order critical point from Proposition 5 (since all second-order critical points are also first-order critical points).

## D    Proofs of sublemmas for local minimality (Lemma 1)

### D.1    Further discussion of challenges

In this section, we further discuss challenges associated with proving $\widetilde{Y}$ is a local minimum in a "traditional" way. For example, [WW20], which constructs spurious local minima for (MC-BM) when $p < \sqrt{2n}$, similarly first constructs spurious second-order critical points and then proves they are additionally local minima. However, their proof follows because their spurious second-order critical points are non-degenerate [WW20, Def. 3 and Rem. 1], which corresponds to the rank of the Riemannian Hessian being sufficiently high. (See Remark 2 in that paper.) We show that *every* spurious second-order critical point for (MC-BM) is degenerate when $p$ is above the Barvinok-Pataki bound, meaning this approach won't work:

**Proposition 9** (Spurious second-order critical points are degenerate when $p \geq \sqrt{2n}$)**.** *Let $Y \in \mathcal{M}_p$ be a spurious second-order critical point for an (arbitrary) instance of* (MC-BM) *where $\frac{p(p+1)}{2} > n$. Then $Y$ is degenerate* [WW20, Def. 3 and Rem. 1]*.*

*Proof.* Theorem 1.6 from [BVB18] gives that any spurious second-order critical point for (MC-BM) must be full rank, meaning $\operatorname{rank}(Y) = p$. Let $S := A - \operatorname{diag}(\nu)$, where $\nu$ is defined as in (1). Then the first-order criticality of $Y$ (Proposition 2) implies $SY = 0$, meaning $\operatorname{rank}(S) \leq n - p$. We have (see Section 2.1):

$$
\begin{aligned}
\langle \operatorname{Hess} \mathrm{OBJ}(Y)[U], U \rangle &= 2 \left\langle S, UU^\top \right\rangle \\
&= 2 \operatorname{vec}(U)^\top \operatorname{vec}(SUI_p) \\
&= 2 \operatorname{vec}(U)^\top (I_p \otimes S) \operatorname{vec}(U),
\end{aligned}
\tag{15}
$$

for any $U \in \mathrm{T}_Y \mathcal{M}_p$. Here, $\operatorname{vec}(Z)$ stacks the columns of $Z$ on top of one another to convert $Z$ into a column vector, and $\otimes$ denotes the Kronecker product. In (15), we used the fact that $\operatorname{vec}(CXB) = (B^\top \otimes C) \operatorname{vec}(X)$.

Then clearly $\operatorname{rank}(\operatorname{Hess} \mathrm{OBJ}(Y)) \leq \operatorname{rank}(I_p \otimes S) = \operatorname{rank}(I_p) \cdot \operatorname{rank}(S) \leq p(n-p)$. Recall from [WW20, Rem. 1] that $Y$ is degenerate if

$$
\operatorname{rank}(\operatorname{Hess} \mathrm{OBJ}(Y)) < \dim(\mathcal{M}_p) - \frac{p(p-1)}{2} = np - n - \frac{p(p-1)}{2}.
$$

(See, e.g., [BVB18, Proposition 1.2] for the fact that $\dim(\mathcal{M}_p) = np - n$.) Then $Y$ is degenerate if

$$
p(n-p) < np - n - \frac{p(p-1)}{2} \iff \frac{p(p+1)}{2} > n.
$$

$\square$

**Degeneracy of higher-order derivatives at $\widetilde{Y}$.**    We know from Proposition 6 that $\widetilde{Y}$ is a spurious second-order critical point if and only if the cost matrix takes the form

$$
A = \begin{bmatrix} P & P \\ P & P \end{bmatrix} + \operatorname{diag}(\alpha)
\tag{16}
$$

for some $\alpha \in \mathbb{R}^n$ and strictly pseudo-PSD $P \in \mathbb{S}^{\frac{n}{2} \times \frac{n}{2}}$. A natural question is whether higher-order Riemannian derivatives[6] could be used to identify an additional condition under which $\widetilde{Y}$ is a (spurious) local minimum. For example, one may hope to show that when $P$ is additionally (strictly) pseudo-PD (the condition identified in Lemma 1), then the fourth derivative is positive. Unfortunately this is not possible, as one can show that *all* higher-order Riemannian derivatives at $\widetilde{Y}$ are degenerate when the cost matrix takes the form (16).

This follows via an examination of $\mathrm{T}_{\widetilde{Y}} \mathcal{M}_{n/2}$. Indeed, consider the subspace $W \subseteq \mathrm{T}_{\widetilde{Y}} \mathcal{M}_{n/2}$ given by those matrices of the form

$$
W := \left\{ \begin{bmatrix} G \\ -G \end{bmatrix} : G \in \mathbb{S}^{\frac{n}{2} \times \frac{n}{2}}, G_{ii} = 0 \text{ for all } i \in [n/2] \right\}.
\tag{17}
$$

---

[6]See Section 10.7 of [Bou22] for an introduction to higher-order Riemannian derivatives.

It is easily observed that $W$ is contained in the kernel of Hess $\mathrm{OBJ}(\widetilde{Y})$ when the cost matrix takes the form (16). Furthermore, one can show that $W$ is orthogonal to the vertical space at $\widetilde{Y}$ [Bou22, Def. 9.24] when we consider the quotient manifold $\mathcal{M}_{n/2}^{\mathrm{full}}/O(n/2)$, where $\mathcal{M}_{n/2}^{\mathrm{full}}$ denotes the open subset of $\mathcal{M}_{n/2}$ containing its rank $n/2$ elements and $O(n/2)$ denotes the orthogonal group in dimension $n/2$. Indeed, the vertical space at $\widetilde{Y}$ consists of all tangent vectors of the form $\widetilde{Y}B$ where $B \in \mathbb{R}^{\frac{n}{2} \times \frac{n}{2}}$ is skew-symmetric. (See, e.g., p. 6 of [WW20].) Thus, the vertical space at $\widetilde{Y}$ is precisely the subspace of $\mathrm{T}_{\widetilde{Y}}\mathcal{M}_{n/2}$ taking the form

$$\left\{ \begin{bmatrix} H \\ -H \end{bmatrix} : H \in \mathbb{R}^{\frac{n}{2} \times \frac{n}{2}}, H \text{ is skew-symmetric} \right\},$$

which is clearly orthogonal to $W$.

Since tangent vectors in the subspace $W$ are in the kernel of Hess $\mathrm{OBJ}(\widetilde{Y})$ but not in the vertical space,[7] one may worry that following a smooth curve $c : I \to \mathcal{M}_{n/2}$ ($I$ is an open interval of $\mathbb{R}$ containing 0) such that $c(0) = \widetilde{Y}, c'(0) \in W$ could yield a decrease in the objective value for sufficiently small inputs $t > 0$. Indeed, one must rule out such behavior to prove $\widetilde{Y}$ is a local minimum.

Unfortunately, higher-order Riemannian derivatives at $\widetilde{Y}$ all also contain $W$ in their zero sets, so they cannot a priori be used to rule out this behavior. Recall that formally, the $k$th Riemannian derivative of $\mathrm{OBJ}$ is a tensor field of order $k$ [Bou22, Def. 10.76] given by $\nabla^k \mathrm{OBJ}$, where $\nabla$ denotes the total covariant derivative [Bou22, Def. 10.77]. (Elsewhere in the paper we have used $\nabla$ to denote the classical Euclidean derivative, but the usage of $\nabla$ in this section is different; the total covariant derivative is *not* (in general) the Euclidean derivative.) Furthermore, recall that tensor fields are pointwise objects, so we use the notation $\nabla^k \mathrm{OBJ}(Y) : \underbrace{\mathrm{T}_Y \mathcal{M}_{n/2} \times \cdots \times \mathrm{T}_Y \mathcal{M}_{n/2}}_{k \text{ times}} \to \mathbb{R}$ to denote

the $k$-linear function associated to a point $Y \in \mathcal{M}_{n/2}$. Then we have the following result:

**Proposition 10** (Higher-order Riemannian derivatives at $\widetilde{Y}$ are degenerate). *For* (MC-BM) *when* $p = n/2$, *suppose the cost matrix takes the form*

$$A = \begin{bmatrix} B & B \\ B & B \end{bmatrix} + \mathrm{diag}(\alpha) \tag{18}$$

*for some $\alpha \in \mathbb{R}^n$ and $B \in \mathbb{S}^{\frac{n}{2} \times \frac{n}{2}}$. Then the subspace $W \subseteq \mathrm{T}_{\widetilde{Y}}\mathcal{M}_{n/2}$ defined in* (17) *is in the zero set of $\nabla^k \mathrm{OBJ}(\widetilde{Y})$ for all $k \geq 1$. (By this we mean $\nabla^k \mathrm{OBJ}(\widetilde{Y})(U, \ldots, U) = 0$ for any $U \in W$).*

Note that Proposition 10 of course additionally encompasses the cases where $B$ is pseudo-PSD, pseudo-PD, etc.

*Proof.* For notational brevity, let $p$ be shorthand for $n/2$ in this proof. Let $c : I \to \mathcal{M}_p$ be a geodesic [Bou22, Def. 5.38] such that $c(0) = \widetilde{Y}, c'(0) = U \in W$. ($I$ is an open interval in $\mathbb{R}$ containing 0.) A simple extension of Example 10.81 in [Bou22] implies

$$\nabla^k \mathrm{OBJ}(\widetilde{Y})(U, \cdots, U) = (\mathrm{OBJ} \circ c)^{(k)}(0)$$

for any $k \geq 1$. (Here, note that $\mathrm{OBJ} \circ c : I \to \mathbb{R}$, so $(\mathrm{OBJ} \circ c)^{(k)}$ is the "usual" $k$th derivative of a function from (an open interval in) $\mathbb{R}$ to $\mathbb{R}$.) Thus, it is sufficient to exhibit such a geodesic $c$ such that $\mathrm{OBJ} \circ c$ is constant over $I$ (implying $(\mathrm{OBJ} \circ c)^{(k)}(0) = 0$).

To construct this geodesic, we will take advantage of the fact that $\mathcal{M}_p$ is a product manifold formed by the Cartesian product of $n$ unit spheres in $\mathbb{R}^p$. Recall that for the unit sphere $\mathrm{S}^{p-1}$ with $x \in \mathrm{S}^{p-1}, v \in \mathrm{T}_x \mathrm{S}^{p-1}$, the curve

$$z_{x,v}(t) = \cos(t\|v\|)x + \frac{\sin(t\|v\|)}{\|v\|}v$$

---

[7]In fact, one can show that when $P$ in (16) is pseudo-PD, then $W$ contains *all* tangent vectors which are in the kernel of Hess $\mathrm{OBJ}(\widetilde{Y})$ but not in the vertical space.

(with the usual smooth extension $\sin(t)/t = 0$ at $t = 0$) is a geodesic which traces the great circle on the sphere from $x$ in the direction $v$. (See Example 5.37 in [Bou22].) Of course, $z'_{x,v}(0) = v$.

Then, viewing $\mathcal{M}_p$ in the form $(\underbrace{\mathbb{S}^{p-1}, \dots, \mathbb{S}^{p-1}}_{n \text{ times}})$ with the $i$th entry corresponding to the $i$th row in $\mathcal{M}_p$, we choose $c(t) = \left( z_{\widetilde{Y}_1, U_1}(t), \dots, z_{\widetilde{Y}_n, U_n}(t) \right)$, where $\widetilde{Y}_i, U_i \in \mathbb{R}^p$ denote the $i$th rows of $\widetilde{Y}, U$ (taken as column vectors). Then $c(t)$ is a geodesic (e.g., [Bou22, Exerc. 5.39]) and $c(0) = \widetilde{Y}, c'(0) = U$. All that is left is to show that OBJ $\circ$ $c$ is constant. This follows due to the form of $W$; it is easy to check that for all sufficiently small $t$, we have that $c(t)$ is an *antipodal configuration* as defined in Section 5. (Recall that antipodal configurations take the form $\begin{bmatrix} G \\ -G \end{bmatrix} \in \mathcal{M}_{n/2}$ for some $G \in \mathbb{R}^{\frac{n}{2} \times \frac{n}{2}}$.) And it is easy to see that when the cost matrix takes the form (18), all antipodal configurations have the same objective value. $\qquad\square$

Thus, we have identified (a subspace of) tangent vectors at $\widetilde{Y}$ which are not in the vertical space at $\widetilde{Y}$ and which lie in the zero sets of all higher-order Riemannian derivatives at $\widetilde{Y}$. As a result, it is not clear how higher-order Riemannian derivatives at $\widetilde{Y}$ can be used to prove $\widetilde{Y}$ is a local minimum.

### D.2 Proof of Lemma 3

Taken together, the proof of Lemma 3 is by far the longest in this paper and will itself utilize several sublemmas given in this section (with some additional very minor claims proven in Section D.4). See the very end of this section for the proof of Lemma 3 itself.

**Important setup for this section.** Throughout Section D.2, we assume we are in the setting of Lemma 1 with pseudo-PD $M \in \mathbb{S}^{\frac{n}{2} \times \frac{n}{2}}$. Furthermore, we assume for simplicity that $\alpha = 0$. This is without loss of generality because due to the feasibility constraint of (MC-BM), shifting the diagonal entries of the cost matrix just corresponds to adding the same constant to the objective value of each feasible point. In particular, it is easy to check that changing $\alpha$ does not affect the geometry of the problem, i.e., the Riemannian derivatives at any point $Y \in \mathcal{M}_p$ remain unchanged. As a result of these assumptions, the cost matrix in this section always takes the form

$$A = \begin{bmatrix} M & M \\ M & M \end{bmatrix} \tag{19}$$

for some pseudo-PD $M \in \mathbb{S}^{\frac{n}{2} \times \frac{n}{2}}$. Finally, $p$ is always $n/2$ in this section. (We may sometimes write $p$ instead of $n/2$ for shorthand.)

To start, the following sublemma, which was described briefly in words in Section 5, provides the backbone of the argument. Recall once again that by a neighborhood of $Y \in \mathcal{M}_p$, we mean a set of the form $\{Y' \in \mathcal{M}_p : ||Y - Y'|| < \epsilon\}$ for some $\epsilon > 0$, where $|| \cdot ||$ as always denotes the Euclidean (or equivalently Frobenius) norm.

**Lemma 5** (Decrease in the potential $\Phi$). *Let the potential $\Phi : \mathcal{M}_{n/2} \to \mathbb{R}_{\geq 0}$ be defined as* $\Phi\left( \begin{bmatrix} G_1 \\ G_2 \end{bmatrix} \right) := ||G_1 + G_2||^2$, *where $G_1, G_2 \in \mathbb{R}^{\frac{n}{2} \times \frac{n}{2}}$. Then there exists a neighborhood $O$ of $\widetilde{Y}$ and $\bar{\eta} > 0$ such that for any $Y \in O$ and $\eta < \bar{\eta}$, we have*

$$\Phi(Y'') \leq (1 - \eta K)\,\Phi(Y). \tag{20}$$

*Here, $Y'' \in \mathcal{M}_{n/2}$ is the point reached by a single step of Riemannian gradient descent starting from $Y$ with step size $\eta$. (The notation $Y''$ is used as $Y'$ is reserved for something else in the proof.) $K > 0$ is a constant which depends only on the instance of (MC-BM).*

*Proof.* We first provide a brief overview of the proof and introduce some notation. To begin, we will represent $Y$ explicitly in the form $Y = \widetilde{Y} + \Delta \in \mathcal{M}_{n/2}$, where $\Delta \in \mathbb{R}^{n \times p}$ should be thought of as a perturbation matrix. (Recall that $p$ is always $n/2$ in this section and may be used as a shorthand.) Using this representation, we derive explicit expressions for $Y'$ and then $\Phi(Y')$,

| Notation | Description |
|---|---|
| $p$ | $p = n/2$ always for this section and may be used as a shorthand |
| $A \in \mathbb{S}^{n \times n}$ | cost matrix taking the form (19) |
| $M \in \mathbb{S}^{p \times p}$ | one pseudo-PD block of the cost matrix; see (19) |
| $\Phi$ | potential; see Lemma 5 |
| $\widetilde{Y}$ | the axial position as in Definition 2; in matrix block form: $I_p$ over $-I_p$ |
| $Y \in \mathcal{M}_p$ | $\widetilde{Y} + \Delta$ (an arbitrary point near $\widetilde{Y}$) |
| $\Delta \in \mathbb{R}^{n \times p}$ | perturbation matrix used to define $Y$; see the line above |
| $Y' \in \mathbb{R}^{n \times p}$ | $Y - \eta \mathrm{grad}\, \mathrm{OBJ}(Y)$ (the point we get to with a gradient step from $Y$) |
| $Y'' \in \mathcal{M}_p$ | the retracted (row-normalized) $Y'$ (equivalently the result of taking a single step of Riemannian gradient descent from $Y$) |
| $Z_i$ | used to denote the $i$th row of $Z$ (taken as a column vector) for a given matrix $Z$ |
| $i^+, i^-$ | $i$ and $i + p$ resp. for $i \in [p]$ |
| $\mathcal{E}, \mathcal{E}' \in \mathbb{R}^{p \times p}$ | defined via their rows: $\mathcal{E}_i = Y_{i^+} + Y_{i^-} = \Delta_{i^+} + \Delta_{i^-}$ and $\mathcal{E}_i' = Y'_{i^+} + Y'_{i^-}$ for all $i \in [p]$ |
| $\mathcal{E}_{*i \setminus ii} \in \mathbb{R}^{p-1}$ | the $i$th column of $\mathcal{E}$ with its $i$th entry removed (only used once!) |
| $M[i] \in \mathbb{S}^{(p-1) \times (p-1)}$ | the submatrix of $M$ formed by removing its $i$th row and column |
| $\lambda_{\min}(\cdot)$ | minimum eigenvalue of the input |
| $\mu_{\mathrm{low}}$ | $\min_{\ell \in [p]} \lambda_{\min}(M[\ell])$ |
| $e_i$ | the $i$th standard basis vector |
| $\|\cdot\|$ | Euclidean (or equivalently Frobenius) norm |

Figure 3: Notation guide for the proof of Lemma 5

where $Y' := Y - \eta \mathrm{grad}\, \mathrm{OBJ}(Y)$. Recalling the contents of Section 2.3, $Y''$ is $Y'$ followed by a normalization of each row. So $Y'$ takes a gradient step from $Y$ but doesn't normalize the rows, meaning (assuming $\mathrm{grad}\, \mathrm{OBJ}(Y) \neq 0$) $Y' \notin \mathcal{M}_{n/2}$. (Thus, we abuse notation here and extend the domain of $\Phi$ to $\mathbb{R}^{n \times p}$.) That said, it is easy to show $\Phi(Y'') \leq \Phi(Y')$, so bounding $\Phi(Y')$ is sufficient. We are then able to bound $\Phi(Y')$ by the right-hand side of (20) by taking the step size and $\|\Delta\|$ to be sufficiently small and using the pseudo-PD property of $M$.

We now delve into the technical details. We will unfortunately need to introduce a significant amount of notation as we go since we will be performing the above analysis in a row-wise manner. (Which is natural in some sense when we recall that $\mathcal{M}_p$ is a product manifold formed by taking the Cartesian product of $n$ unit spheres in $\mathbb{R}^p$. And for product manifolds, geometric entities such as the tangent space and Riemannian derivatives can be expressed as products or concatenations of entities over the constituent manifolds.) As an aid, Figure 3 can be used as a reference for the notation used in this proof.

Letting $i^+, i^-$ denote $i, i + p$ respectively for $i \in [p]$, we first derive expressions for $Y'_{i^+}, Y'_{i^-} \in \mathbb{R}^p$ for all $i \in [p]$. Recall that $Y' = Y - \eta \mathrm{grad}\, \mathrm{OBJ}(Y) = Y - 2\eta(A - \mathrm{diag}(\nu))Y$ with $\nu$ defined as in (1). Then

$$Y'_{i^+} = Y_{i^+} - 2\eta \left( \sum_{j=1}^{n} A_{ij} Y_j - \sum_{j=1}^{n} A_{ij} \langle Y_{i^+}, Y_j \rangle Y_{i^+} \right)$$

$$= Y_{i^+} - 2\eta \left( \sum_{j=1}^{p} M_{ij}(Y_{j^+} + Y_{j^-}) - \sum_{j=1}^{p} M_{ij} \langle Y_{i^+}, Y_{j^+} + Y_{j^-} \rangle Y_{i^+} \right)$$

$$= Y_{i^+} - 2\eta \sum_{j=1}^{p} M_{ij} \left( \mathcal{E}_j - \langle Y_{i^+}, \mathcal{E}_j \rangle Y_{i^+} \right).$$

The second line uses the block form of $A$, and the third line introduces new notation: we let $\mathcal{E} \in \mathbb{R}^{p \times p}$ be defined such that the $i$th row of $\mathcal{E}$ is $\mathcal{E}_i = Y_{i+} + Y_{i-} = \Delta_{i+} + \Delta_{i-}$ for $i \in [p]$. The matrix $\mathcal{E}$ is directly related to the potential $\Phi$; indeed, $\Phi(Y) = ||\mathcal{E}||^2$.

Similarly, one can derive

$$Y_{i-}' = Y_{i-} - 2\eta \sum_{j=1}^{p} M_{ij} \left( \mathcal{E}_j - \langle Y_{i-}, \mathcal{E}_j \rangle Y_{i-} \right).$$

Next, we define $\mathcal{E}' \in \mathbb{R}^{p \times p}$ analogously to $\mathcal{E}$ but using $Y'$: the $i$th row of $\mathcal{E}'$ is $\mathcal{E}_i' = Y_{i+}' + Y_{i-}'$ for $i \in [p]$. Thus, $\Phi(Y') = ||\mathcal{E}'||^2$. (We abuse notation and extend the domain of $\Phi$ to $\mathbb{R}^{n \times p}$.) We will bound $\Phi(Y')$ through $\mathcal{E}'$ in a row-wise manner. Using the expressions we have derived, we have for $i \in [p]$:

$$\mathcal{E}_i' = Y_{i+}' + Y_{i-}'$$
$$= \mathcal{E}_i + 2\eta \sum_{j=1}^{p} M_{ij} \left[ \langle Y_{i+}, \mathcal{E}_j \rangle Y_{i+} + \langle Y_{i-}, \mathcal{E}_j \rangle Y_{i-} - 2\mathcal{E}_j \right].$$

Then for $i \in [p]$,

$$\|\mathcal{E}_i'\|^2 = \sum_{\ell=1}^{p} \langle \mathcal{E}_i', e_\ell \rangle^2$$

$$= \sum_{\ell=1}^{p} \left[ \langle \mathcal{E}_i, e_\ell \rangle + 2\eta \sum_{j=1}^{p} M_{ij} \left[ \langle Y_{i+}, \mathcal{E}_j \rangle \langle Y_{i+}, e_\ell \rangle + \langle Y_{i-}, \mathcal{E}_j \rangle \langle Y_{i-}, e_\ell \rangle - 2 \langle \mathcal{E}_j, e_\ell \rangle \right] \right]^2$$

$$= \sum_{\ell=1}^{p} \left[ \langle \mathcal{E}_i, e_\ell \rangle^2 + O(\eta^2) \right.$$

$$\left. + 2\eta \sum_{j=1}^{p} M_{ij} \left[ \langle \mathcal{E}_i, e_\ell \rangle \left( \langle Y_{i+}, \mathcal{E}_j \rangle \langle Y_{i+}, e_\ell \rangle + \langle Y_{i-}, \mathcal{E}_j \rangle \langle Y_{i-}, e_\ell \rangle \right) - 2 \langle \mathcal{E}_i, e_\ell \rangle \langle \mathcal{E}_j, e_\ell \rangle \right] \right]$$

$$= \|\mathcal{E}_i\|^2 + O(\eta^2)$$

$$+ 2\eta \sum_{\ell=1}^{p} \sum_{j=1}^{p} M_{ij} \left[ \langle \mathcal{E}_i, e_\ell \rangle \left( \langle Y_{i+}, \mathcal{E}_j \rangle \langle Y_{i+}, e_\ell \rangle + \langle Y_{i-}, \mathcal{E}_j \rangle \langle Y_{i-}, e_\ell \rangle \right) - 2 \langle \mathcal{E}_i, e_\ell \rangle \langle \mathcal{E}_j, e_\ell \rangle \right].$$

The $O(\eta^2)$ hides terms that depend on the perturbation $\Delta$, but this will not matter as $\eta$ will be taken sufficiently small in the final step after a bound on $\|\Delta\|$ is set. (The $O(\eta^2)$ also hides terms that depend on the instance of (MC-BM), but these do not matter for our purposes.)

Then

$$\Phi(Y') = \|\mathcal{E}'\|^2$$

$$= \sum_{i=1}^{p} \|\mathcal{E}_i'\|^2$$

$$= \|\mathcal{E}\|^2 + O(\eta^2)$$

$$+ 2\eta \underbrace{\sum_{i=1}^{p} \sum_{\ell=1}^{p} \sum_{j=1}^{p} M_{ij} \left[ \langle \mathcal{E}_i, e_\ell \rangle \left( \langle Y_{i+}, \mathcal{E}_j \rangle \langle Y_{i+}, e_\ell \rangle + \langle Y_{i-}, \mathcal{E}_j \rangle \langle Y_{i-}, e_\ell \rangle \right) \right]}_{\text{①}} \qquad (21)$$

$$- 4\eta \underbrace{\sum_{i=1}^{p} \sum_{\ell=1}^{p} \sum_{j=1}^{p} M_{ij} \langle \mathcal{E}_i, e_\ell \rangle \langle \mathcal{E}_j, e_\ell \rangle}_{\text{②}}.$$

We now upper bound ① and lower bound ② starting with the former, which relies on the key observation that when $i = \ell$, then $\langle \mathcal{E}_i, e_\ell \rangle = \langle \mathcal{E}_i, e_i \rangle$ is small, and when $i \neq \ell$, then $\langle Y_{i+}, e_\ell \rangle$ and $\langle Y_{i-}, e_\ell \rangle$ are small. Formally,

$$
① = \sum_{i=1}^{p} \sum_{\substack{\ell=1 \\ \ell \neq i}}^{p} \sum_{j=1}^{p} M_{ij} \left[ \langle \mathcal{E}_i, e_\ell \rangle \left( \langle Y_{i+}, \mathcal{E}_j \rangle \langle Y_{i+}, e_\ell \rangle + \langle Y_{i-}, \mathcal{E}_j \rangle \langle Y_{i-}, e_\ell \rangle \right) \right]
$$

$$
+ \sum_{i=1}^{p} \sum_{j=1}^{p} M_{ij} \left[ \langle \mathcal{E}_i, e_i \rangle \left( \langle Y_{i+}, \mathcal{E}_j \rangle \langle Y_{i+}, e_i \rangle + \langle Y_{i-}, \mathcal{E}_j \rangle \langle Y_{i-}, e_i \rangle \right) \right]
$$

$$
\leq \sum_{i=1}^{p} \sum_{\substack{\ell=1 \\ \ell \neq i}}^{p} \sum_{j=1}^{p} M_{ij} \left[ \|\mathcal{E}_i\| \left( \|\mathcal{E}_j\| \|\Delta_{i+}\| + \|\mathcal{E}_j\| \|\Delta_{i-}\| \right) \right] \tag{22}
$$

$$
+ \sum_{i=1}^{p} \sum_{j=1}^{p} M_{ij} \left[ \|\mathcal{E}_i\| \|\Delta_{i+} - \Delta_{i-}\| \left( \|\mathcal{E}_j\| + \|\mathcal{E}_j\| \right) \right] \tag{23}
$$

$$
= O(\|\mathcal{E}\|^2 \|\Delta\|). \tag{24}
$$

(22) uses Cauchy-Schwarz (recall that $Y_{i+}, Y_{i-}$ are unit vectors by definition) and the fact that $\langle Y_{i+}, e_\ell \rangle = \langle \Delta_{i+}, e_\ell \rangle \leq \|\Delta_{i+}\|$ since $Y_{i+} = e_i + \Delta_{i+}$ by definition and $i \neq \ell$. (And $\langle Y_{i-}, e_\ell \rangle$ can be bounded similarly.) (23) uses Cauchy-Schwarz as well as the following key bound:

$$
\langle \mathcal{E}_i, e_i \rangle = \langle \Delta_{i+}, e_i \rangle - \langle \Delta_{i-}, -e_i \rangle
$$
$$
= \frac{-\|\Delta_{i+}\|^2 + \|\Delta_{i-}\|^2}{2} \tag{25}
$$
$$
\leq \|\Delta_{i+} + \Delta_{i-}\| \|\Delta_{i+} - \Delta_{i-}\|
$$
$$
= \|\mathcal{E}_i\| \|\Delta_{i+} - \Delta_{i-}\|,
$$

where we have used Lemmas 8 and 9 from Section D.4. This bound is critical; a less tight bound would not work because $\|\mathcal{E}\|$ may be much smaller than $\|\Delta\|$. The big $O$ notation in line (24) hides terms which depend on the instance of (MC-BM), but these don't matter for our purposes.

We now turn our focus to lower bounding ②, which is the only place where we use the fact that $M$ is pseudo-PD. We have

$$
② = \sum_{\ell=1}^{p} \sum_{\substack{i=1 \\ i \neq \ell}}^{p} \sum_{\substack{j=1 \\ j \neq \ell}}^{p} M_{ij} \langle \mathcal{E}_i, e_\ell \rangle \langle \mathcal{E}_j, e_\ell \rangle
$$

$$
+ \sum_{\ell=1}^{p} \sum_{j=1}^{p} M_{\ell j} \langle \mathcal{E}_\ell, e_\ell \rangle \langle \mathcal{E}_j, e_\ell \rangle + \sum_{\ell=1}^{p} \sum_{i=1}^{p} M_{i\ell} \langle \mathcal{E}_i, e_\ell \rangle \langle \mathcal{E}_\ell, e_\ell \rangle - \sum_{\ell=1}^{p} M_{\ell\ell} \langle \mathcal{E}_\ell, e_\ell \rangle \langle \mathcal{E}_\ell, e_\ell \rangle
$$

$$
\geq -O(\|\mathcal{E}\|^2 \|\Delta\|) + \sum_{\ell=1}^{p} \sum_{\substack{i=1 \\ i \neq \ell}}^{p} \sum_{\substack{j=1 \\ j \neq \ell}}^{p} M_{ij} \mathcal{E}_{i\ell} \mathcal{E}_{j\ell} \tag{26}
$$

$$
= -O(\|\mathcal{E}\|^2 \|\Delta\|) + \sum_{\ell=1}^{p} \mathcal{E}_{*\ell \setminus \ell\ell}^{\top} M[\ell] \mathcal{E}_{*\ell \setminus \ell\ell} \tag{27}
$$

$$
\geq -O(\|\mathcal{E}\|^2 \|\Delta\|) + \sum_{\ell=1}^{p} \mu_{\text{low}} \|\mathcal{E}_{*\ell \setminus \ell\ell}\|^2 \tag{28}
$$

$$
= -O(\|\mathcal{E}\|^2 \|\Delta\|) + \mu_{\text{low}} \left( \|\mathcal{E}\|^2 - \sum_{i=1}^{p} \mathcal{E}_{ii}^2 \right) \tag{29}
$$

$$
= -O(\|\mathcal{E}\|^2 \|\Delta\|) + \mu_{\text{low}} \left( \|\mathcal{E}\|^2 - \sum_{i=1}^{p} \langle \mathcal{E}_i, e_i \rangle^2 \right) \tag{30}
$$

$$\geq -O(\|\mathcal{E}\|^2 \|\Delta\|) + \mu_{\text{low}} \left( \|\mathcal{E}\|^2 - \sum_{i=1}^{p} \|\mathcal{E}_i\|^2 \|2\Delta\|^2 \right) \tag{31}$$

$$= -O(\|\mathcal{E}\|^2 \|\Delta\|) + \mu_{\text{low}} \left( 1 - \|2\Delta\|^2 \right) \|\mathcal{E}\|^2 \tag{32}$$

(26) once again uses the key inequality (25) (and Cauchy-Schwarz) and also simply rewrote $\langle \mathcal{E}_i, e_\ell \rangle = \mathcal{E}_{i\ell}, \langle \mathcal{E}_j, e_\ell \rangle = \mathcal{E}_{j\ell}$. (27) introduces the unfortunate notation $\mathcal{E}_{*\ell \backslash \ell\ell} \in \mathbb{R}^{p-1}$, which denotes the $\ell$th column of $\mathcal{E}$ except the $\ell$th entry of this column (aka $\mathcal{E}_{\ell\ell}$) has been removed. Recall that $M[\ell] \in \mathbb{S}^{(p-1)\times(p-1)}$ as always denotes the submatrix of $M$ formed by removing the $\ell$th row and column. Then, (27) follows by observing that the inner two summations on the right side of (26) form a quadratic form which is precisely $\mathcal{E}_{*\ell \backslash \ell\ell}^\top M[\ell] \mathcal{E}_{*\ell \backslash \ell\ell}$. In (28), we introduce the notation $\mu_{\text{low}}$ which denotes $\min_{\ell \in [p]} \lambda_{\min}(M[\ell])$, where $\lambda_{\min}(\cdot)$ denotes the minimum eigenvalue of its argument. In other words, $\mu_{\text{low}}$ lower bounds the eigenvalues of $M[\ell]$ for any $\ell$, and the fact that $\mu_{\text{low}} > 0$ follows from the fact that $M$ is pseudo-PD. (29) follows by expanding $\sum_{\ell=1}^{p} \left\| \mathcal{E}_{*\ell \backslash \ell\ell} \right\|^2$ and noting that only the diagonal entries of $\mathcal{E}$ are not covered. In (30) we simply rewrite $\mathcal{E}_{ii} = \langle \mathcal{E}_i, e_i \rangle$, and (31) uses the key inequality (25). (32) simply uses $\sum_{i=1}^{p} \|\mathcal{E}_i\|^2 = \|\mathcal{E}\|^2$.

Now going back to (21) and using the bounds on ① and ②, we have

$$\Phi(Y') \leq \|\mathcal{E}\|^2 - \eta \mu_{\text{low}} \left( 1 - \|2\Delta\|^2 \right) \|\mathcal{E}\|^2 + \eta O(\|\mathcal{E}\|^2 \|\Delta\|) + O(\eta^2)$$

$$\leq (1 - \eta K) \|\mathcal{E}\|^2,$$

where in the last line we took $\|\Delta\|$ and then $\eta$ to be sufficiently small. Finally, recall that $\Phi(Y) = \|\mathcal{E}\|^2$, and note also that the norm of each row of $Y'$ is at least 1.[8] Then Lemma 10 from Section D.4 implies $\Phi(Y'') \leq \Phi(Y')$, and we are done. $\qquad \square$

Next, we would like to extend the result of Lemma 5 to all consecutive pairs of iterates produced by Riemannian gradient descent and not just the first pair. This will be done shortly in Lemma 7, but in preparation we first show that as long as the iterates of Riemannian gradient descent stay in the neighborhood $O$ identified in Lemma 5, they form a Cauchy sequence where the distances between consecutive iterates decrease geometrically.

**Lemma 6** (Iterates confined to neighborhood form Cauchy sequence). *Let $O, \bar{\eta}, K$ denote the neighborhood, step-size bound, and constant identified in Lemma 5. Suppose Riemannian gradient descent with step size $\eta < \bar{\eta}$ is initialized at some $Y^{(0)} \in O$, and furthermore $Y^{(1)}, \ldots, Y^{(t)} \in O$. Then for any $k \in \{0, \ldots, t\}$, we have*

$$\|Y^{(k)} - Y^{(k+1)}\| \leq 4\eta \|M\| \|Y^{(0)} - \widetilde{Y}\|(1 - \eta K)^{k/2}.$$

Before starting the proof, note that we critically do not require $Y^{(t+1)} \in O$. This will be important in the proof of Lemma 7.

*Proof.* We have for $k \in \{0, \ldots, t\}$:

$$\left\| Y^{(k)} - Y^{(k+1)} \right\|^2 = \left\| Y^{(k)} - \text{R}_{Y^{(k)}} \left( -\eta \text{grad OBJ}(Y^{(k)}) \right) \right\|^2 .$$

Now, recall from Section 2.3 that $\text{R}_{Y^{(k)}} \left( -\eta \text{grad OBJ}(Y^{(k)}) \right)$ is equal to $Y^{(k)} - \eta \text{grad OBJ}(Y^{(k)})$ followed by a normalization of each row. We claim

$$\left\| Y^{(k)} - \text{R}_{Y^{(k)}} \left( -\eta \text{grad OBJ}(Y^{(k)}) \right) \right\|^2 \leq \left\| Y^{(k)} - \left( Y^{(k)} - \eta \text{grad OBJ}(Y^{(k)}) \right) \right\|^2$$

$$= \left\| \eta \text{grad OBJ}(Y^{(k)}) \right\|^2 .$$

Indeed, this follows from applying Lemma 11 from Section D.4 in a row-wise manner, where $u$ is chosen as the $i$th row of $Y^{(k)}$, $v$ is chosen as the $i$th row of $\text{R}_{Y^{(k)}} \left( -\eta \text{grad OBJ}(Y^{(k)}) \right)$, $\alpha = 1$, and

---

[8]This follows because $Y'$ takes the form of "a point on $\mathcal{M}_{n/2}$ plus a tangent vector," and because tangent vectors are row-wise orthogonal to their base, clearly adding one can only increase the norm of each row.

$\beta$ is set to the norm of the $i$th row of $Y^{(k)} - \eta \text{grad} \, \text{OBJ}(Y^{(k)})$. ($\beta \geq 1$ since $\eta \text{grad} \, \text{OBJ}(Y^{(k)}$ is row-wise orthogonal to $Y^{(k)}$ due to the former being a tangent vector.) Intuitively, we are just using the fact that retraction onto the sphere can only bring you closer to your starting point.

Next, note that

$$\left\| \eta \text{grad} \, \text{OBJ}(Y^{(k)}) \right\|^2 \leq \left\| \eta \nabla \text{OBJ}(Y^{(k)}) \right\|^2 = \left\| 2\eta A Y^{(k)} \right\|^2,$$

where $\nabla$ is used to denote the classical Euclidean gradient. This follows because, as mentioned in Section 2.1, the Riemannian gradient is equal to the Euclidean gradient composed with an orthogonal projection, and an orthogonal projection can only decrease the norm. Now write $Y^{(k)} = \begin{bmatrix} G_1 \\ G_2 \end{bmatrix}$ where $G_1, G_2 \in \mathbb{R}^{\frac{n}{2} \times \frac{n}{2}}$. Using the form of $A$ given in (19), it is easy to see that

$$\| 2\eta A Y^{(k)} \|^2 = \| 4\eta M (G_1 + G_2) \|^2 \leq 16\eta^2 \| M \|^2 \| G_1 + G_2 \|^2,$$

where we used the fact that the Frobenius norm is submultiplicative. Then, note that by definition, $\Phi(Y^{(k)}) = \| G_1 + G_2 \|^2$. Connecting the dots, we have shown up to this point that

$$\| Y^{(k)} - Y^{(k+1)} \|^2 \leq 16\eta^2 \| M \|^2 \Phi(Y^{(k)}). \tag{33}$$

Using Lemma 5 and the fact that the iterates up to step $k$ are in $O$, we have $\Phi(Y^{(k)}) \leq (1 - \eta K)^k \Phi(Y^{(0)})$. (Note that when $k = t$, we are critically relying on the fact that Lemma 5 does not require $Y'' \in O$.) Furthermore, it is easy to see that $\Phi$ always bounds the distance away from $\widetilde{Y}$; indeed, let $Z = \begin{bmatrix} Z_1 \\ Z_2 \end{bmatrix}$ with $Z_1, Z_2 \in \mathbb{R}^{\frac{n}{2} \times \frac{n}{2}}$ and note that

$$\begin{aligned} \Phi(Z) &= \| Z_1 + Z_2 \|^2 \\ &= \| Z_1 + I + Z_2 - I \|^2 \\ &\leq \| Z_1 + I \|^2 + \| Z_2 - I \|^2 \\ &= \| Z - \widetilde{Y} \|^2. \end{aligned}$$

(33) and the contents of the previous paragraph imply the desired result. $\qquad\square$

Now we finally extend Lemma 5 to all consecutive pairs of iterates produced by Riemannian gradient descent. This can of course be achieved by confining all iterates to the neighborhood identified in Lemma 5. The following lemma does this by initializing in an even smaller neighborhood of $\widetilde{Y}$.

**Lemma 7** (Riemannian gradient descent stays close to $\widetilde{Y}$). *Let $O, \bar{\eta}, K$ denote the neighborhood, step-size bound, and constant identified in Lemma 5. Then there exists a neighborhood $S \subseteq O$ of $\widetilde{Y}$ such that if Riemannian gradient descent with step size $\eta < \min\left\{ \bar{\eta}, \frac{1}{2K} \right\}$ is initialized at any $Y^{(0)} \in S$, all future iterates lie in $O$.*

*Proof.* We proceed by induction on the iterate counter $t$, and will choose $S$ in the inductive step so that the proof goes through. $Y^{(0)} \in O$ since $S \subseteq O$. Now suppose $Y^{(0)}, \ldots, Y^{(t)} \in O$, and we will set $S$ independently of $t$ so that $Y^{(t+1)} \in O$. We have

$$\begin{aligned} \| \widetilde{Y} - Y^{(t+1)} \| &\leq \| \widetilde{Y} - Y^{(0)} \| + \sum_{k=0}^{t} \| Y^{(k)} - Y^{(k+1)} \| \\ &\leq \| \widetilde{Y} - Y^{(0)} \| + 4\eta \| M \| \| \widetilde{Y} - Y^{(0)} \| \sum_{k=0}^{t} (1 - \eta K)^{k/2} \\ &\leq \| \widetilde{Y} - Y^{(0)} \| \left( 1 + 4\eta \| M \| \sum_{k=0}^{\infty} (1 - \eta K)^{k/2} \right), \end{aligned}$$

where we used Lemma 6 (which critically doesn't require $Y^{(t+1)} \in O$).

Clearly $\left( 1 + 4\eta \| M \| \sum_{k=0}^{\infty} (1 - \eta K)^{k/2} \right)$ is bounded and doesn't depend on $t$, so $\| \widetilde{Y} - Y^{(0)} \|$ can be chosen sufficiently small so that $Y^{(t+1)} \in O$. $\qquad\square$

Finally, we give the proof of Lemma 3:

*Proof of Lemma 3.* We choose $N$ to be the neighborhood $S$ identified in Lemma 7 and pick $\eta' = \min\left\{\bar{\eta}, \frac{1}{2K}\right\}$ with $\bar{\eta}, K$ defined as in Lemma 5. Then due to Lemma 7, all iterates of Riemannian gradient descent with any step size $\eta < \eta'$ initialized at any $Y^{(0)} \in N$ lie in $O$, the neighborhood identified in Lemma 5. As a result, Lemma 6 implies that the sequence of iterates $Y^{(0)}, Y^{(1)}, \ldots$ forms a Cauchy sequence, and since $\mathbb{R}^{n \times p}$ is complete, the iterates converge to some $\overline{Y}$. Clearly $\overline{Y} \in \mathcal{M}_{n/2}$ as all iterates of Riemannian gradient descent lie on $\mathcal{M}_{n/2}$.

Also, Lemma 5 and the fact that all iterates lie in $O$ implies that the sequence $\Phi(Y^{(0)}), \Phi(Y^{(1)}), \ldots$ converges to 0. Since $\Phi$ is continuous, we have $\Phi(\overline{Y}) = 0$. It is easy to see that for $Z \in \mathcal{M}_{n/2}$, we have $\Phi(Z) = 0$ if and only if $Z$ is *antipodal*, where as in Section 5, we say a point is antipodal if it takes the form $\begin{bmatrix} G \\ -G \end{bmatrix} \in \mathcal{M}_{n/2}$ for some $G \in \mathbb{R}^{\frac{n}{2} \times \frac{n}{2}}$. Thus, $\overline{Y}$ is antipodal. Note that $\widetilde{Y}$ is also antipodal, and it is easy to check that all antipodal points have the same objective value. (In particular, if the cost matrix takes the form (19) as we assume without loss of generality in this section, that objective value is 0.) □

## D.3   Proof of Lemma 4

We first restate a result from [Bou22] which yields a quadratic upper bound on the objective in the same style as the classic quadratic upper bound which holds in the Euclidean case when the Euclidean gradient is Lipschitz. (Indeed, the Riemannian gradient $\mathrm{grad}\,\mathrm{OBJ}$ for (MC-BM) is Lipschitz, but defining Lipschitzness for the Riemannian gradient requires care [Bou22, Definition 10.44]. Proposition 11 is sufficient for our purposes since we only need a quadratic upper bound and don't care about the actual value of the Lipschitz constant.)

**Proposition 11** (Quadratic bound [Bou22, Lem. 10.57, abbreviated])**.** *Consider a smooth manifold $\mathcal{M}$, retraction $\mathrm{R}$ on $\mathcal{M}$ [Bou22, Definition 3.47], compact subset $\mathcal{K} \subseteq \mathcal{M}$, and continuous, nonnegative function $r : \mathcal{K} \to \mathbb{R}$. The set*

$$\mathcal{T} = \{(x, s) \in \mathrm{T}\mathcal{M} : x \in \mathcal{K} \text{ and } \|s\| \leq r(x)\}$$

*is compact in the tangent bundle $\mathrm{T}\mathcal{M}$ [Bou22, Definition 3.42]. Assume $f : \mathcal{M} \to \mathbb{R}$ is twice continuously differentiable. Then there exists a constant $L$ such that, for all $(x, s) \in \mathcal{T}$, we have*

$$|f(\mathrm{R}_x(s)) - f(x) - \langle s, \mathrm{grad}\,f(x)\rangle| \leq \frac{L}{2}\|s\|^2.$$

We now give the proof of Lemma 5:

*Proof of Lemma 4.* Formally, our goal is to identify $\tilde{\eta} > 0$ such that for any $\eta < \tilde{\eta}$ and $Y \in \mathcal{M}_p$, we have

$$\mathrm{OBJ}(Y') \leq \mathrm{OBJ}(Y) \quad \text{where } Y' = \mathrm{R}_Y\left(-\eta\mathrm{grad}\,\mathrm{OBJ}(Y)\right),$$

with $\mathrm{R}_Y$ defined as the metric projection retraction for $\mathcal{M}_p$, as in Section 2.3. (In fact, we will show that when $\mathrm{grad}\,\mathrm{OBJ}(Y) \neq 0$, our proof yields a strict decrease: $\mathrm{OBJ}(Y') < \mathrm{OBJ}(Y)$.)

We apply Proposition 11 with $\mathcal{K} \leftarrow \mathcal{M}_p$, as clearly $\mathcal{M}_p$ is compact. The fact that $\mathrm{grad}\,\mathrm{OBJ}$ is continuous implies $\|\mathrm{grad}\,\mathrm{OBJ}(\cdot)\|$ is continuous, and this together with the compactness of $\mathcal{M}_p$ implies there exists some constant $P$ such that $\|\mathrm{grad}\,\mathrm{OBJ}(Z)\| \leq P$ for all $Z \in \mathcal{M}_p$.

Pick $r : \mathcal{M}_p \to \mathbb{R}$ to be the constant function which sends everything to $P$, i.e., $r(Z) = P$ for all $Z \in \mathcal{M}_p$. Then for all $0 < \eta \leq 1$ and $Z \in \mathcal{M}_p$, we have $(Z, -\eta\mathrm{grad}\,\mathrm{OBJ}(Z)) \in \mathcal{T}$.

Then Proposition 11 implies there exists some constant $L$ such that for $0 < \eta \leq 1$, we have

$$\mathrm{OBJ}(Y') - \mathrm{OBJ}(Y) = \mathrm{OBJ}\left(\mathrm{R}_Y\left(-\eta\mathrm{grad}\,\mathrm{OBJ}(Y)\right)\right) - \mathrm{OBJ}(Y)$$

$$\leq \frac{L}{2}\|-\eta\mathrm{grad}\,\mathrm{OBJ}(Y)\|^2 + \langle -\eta\mathrm{grad}\,\mathrm{OBJ}(Y), \mathrm{grad}\,\mathrm{OBJ}(x)\rangle$$

$$= \left(\frac{\eta^2 L}{2} - \eta\right)\|\mathrm{grad}\,\mathrm{OBJ}(Y)\|^2$$

Then $\frac{\eta^2 L}{2} - \eta < 0$ for all $0 < \eta < 2/L$, so setting $\tilde{\eta} \leftarrow \min\left\{1, \frac{1}{L}\right\}$ yields the desired result. □

### D.4 Minor claims

In this section, we prove minor claims that are used in Section D.2.

**Lemma 8** (Close to orthogonal). *Let $v \in \mathbb{R}^m, \delta \in \mathbb{R}^m$ be such that $\|v\| = \|v + \delta\| = 1$. Then $\langle \delta, v \rangle = - \|\delta\|^2 / 2$.*

*Proof.* We have

$$\|v + \delta\|^2 = 1$$
$$\implies \|v\|^2 + 2\langle \delta, v \rangle + \|\delta\|^2 = 1$$
$$\implies \langle \delta, v \rangle = - \|\delta\|^2 / 2.$$

$\square$

**Lemma 9** (Reverse triangle inequality with squares). *Let $u, v \in \mathbb{R}^m$. Then*

$$\left| \|u\|^2 - \|v\|^2 \right| \leq \|u + v\| \, \|u - v\| .$$

*Proof.* Note that

$$\|u\|^2 - \|v\|^2 = \langle u + v, u - v \rangle \leq \|u + v\| \, \|u - v\| .$$

The result follows by symmetry. $\square$

**Lemma 10** (Normalizing doesn't increase the potential). *Set $p = n/2$, and let $\Phi$ be defined as in Lemma 5, although we abuse notation here and extend the domain to $\mathbb{R}^{n \times p}$. Let $Z \in \mathbb{R}^{n \times p}$ be arbitrary except with the single restriction that the norm of each of its rows is at least 1, and let $\overline{Z} \in \mathcal{M}_{n/2}$ denote the matrix formed by normalizing each row of $Z$. Then $\Phi(\overline{Z}) \leq \Phi(Z)$.*

*Proof.* It is clearly sufficient to show that $\left\| \overline{Z}_i + \overline{Z}_{i+p} \right\|^2 \leq \|Z_i + Z_{i+p}\|^2$ for all $i \in [p]$. ($Z_i \in \mathbb{R}^p$ denotes the $i$th row.) This follows from Lemma 11 below. $\square$

**Lemma 11** (Metric projection is contractive). *Let $u, v \in \mathbb{R}^m$ be such that $\|u\| = \|v\| = 1$. Then for any $\alpha, \beta \geq 1$, we have*

$$\|u - v\|^2 \leq \|\alpha u - \beta v\|^2 .$$

*Proof.* We have

$$\begin{aligned}
\|\alpha u - \beta v\|^2 - \|u - v\|^2 &= \alpha^2 + \beta^2 - 2(\alpha\beta - 1)\langle u, v \rangle - 2 \\
&\geq \alpha^2 + \beta^2 - 2(\alpha\beta - 1) - 2 \\
&= \alpha^2 + \beta^2 - 2\alpha\beta \\
&= (\alpha - \beta)^2 \\
&\geq 0.
\end{aligned}$$

In the second line, we used Cauchy-Schwarz and the fact that $\alpha\beta \geq 1$. $\square$

## E  Constructions of strictly pseudo-PD matrices (Lemma 2)

In this section we provide two proofs of Lemma 2, which posits the existence of $k \times k$ strictly pseudo-PD matrices for $k \geq 2$. The first is a probabilistic construction and the second is deterministic. The former has the advantage of having nonnegative entries, which, combined with Lemma 1, results in "natural" cost matrices that can arise as the adjacency matrix of a weighted graph. For the latter deterministic construction, we also characterize the optimal solutions of the associated instances of (MC-SDP), revealing they have a qualitatively different structure than $\widetilde{Y}\widetilde{Y}^\top$.

## E.1 Probabilistic construction with nonnegative entries

Our random construction uses a random matrix $U \in \mathbb{R}^{k \times (k-1)}$ with nonnegative entries such that for each $i \in [k]$, the submatrix $U^{(i)} \in \mathbb{R}^{(k-1) \times (k-1)}$ formed by removing the $i$th row of $U$ has non-negligible least-singular-value (and hence full rank). Each entry of $U$ is generated i.i.d. from a $N(\mu, 1)$ where $\mu = c_0 \sqrt{\log k}$ for a sufficiently large constant $c_0 > 0$. The final matrix is just $M = UU^\top - \varepsilon I$, where $\varepsilon = k^{-\Omega(1)}$ is set appropriately. By construction $UU^\top$ is a $k \times k$ matrix of rank $k-1$; hence $M$ has exactly one negative eigenvalue. In what follows $\lambda_\ell(M)$ denotes the $\ell$th eigenvalue of $M$ (note that $M$ is symmetric in our case; hence all eigenvalues are real).

**Proposition 12** (Randomized construction for Lemma 2). *There exists absolute constants $c_0, c_1, c_2 > 0$ such that the following holds for a given $k \in \mathbb{N}$ with $k \geq 2$. Suppose $U \in \mathbb{R}^{k \times (k-1)}$ is a random matrix where each entry is generated i.i.d. from $N(\mu, 1)$ for $\mu = c_0 \sqrt{\log k}$ and let $M = UU^\top - \varepsilon I$ with $\varepsilon := c_1/k^{c_2}$. Then, with probability at least $1 - 1/k^7$, $M$ is nonnegative and strictly pseudo-PD. In particular,*

*(i) for each $i \in [k]$, the submatrix $M[i] \in \mathbb{S}^{(k-1) \times (k-1)}$ formed by removing the $i$th row and the $i$th column of $M$ satisfies $\lambda_{k-1}(M[i]) \geq \frac{c_1}{k^{c_2}}$.*

*(ii) every entry of $M$ is nonnegative.*

*(iii) it is not positive semidefinite i.e., $\lambda_k(M) \leq -\frac{c_1}{k^{c_2}}$.*

We remark that the constant 7 in the exponent of the failure probability is arbitrarily chosen. We can make this an arbitrarily large constant, and adjust constants $c_0, c_1, c_2 > 0$ appropriately.

The proof of the above lemma using the following claim about the least singular value of square matrices. We remark that much stronger bounds on the least singular values are known in random matrix theory. We state and include a proof (which follows somewhat standard arguments) of the following claim which is more tailored for our needs.

**Lemma 12.** *Fix any $k \in \mathbb{N}$ with $k \geq 1$. Let $A \in \mathbb{R}^{k \times k}$ be a random matrix, each entry of which is sampled $N(\mu, 1)$ for some $\mu$. Then there exists absolute constants $c_1, c_2, c_3 > 0$ such that with probability at least $1 - 1/(k+1)^8$, we have:*

*(i) $s_k(A)^2 > 2c_1/(k+1)^{c_2}$, where $s_k(A)$ denotes the least singular value of $A$.*

*(ii) every entry of $A$ is in the interval $[\mu - c_3\sqrt{\log k}, \mu + c_3\sqrt{\log k}]$.*

*Proof.* We will show separately that both the required properties hold with high probability, and do a union bound over the failure of these two events. Part (ii) of the claim just follows by applying standard Gaussian tail bounds for a fixed entry of the $k \times k$ matrix, and then using a union bound over all the $k^2$ entries.

We now focus on part (i) of the claim. This follows a standard argument using anti-concentration bounds (or small ball probability). Let $a_1, a_2, \ldots, a_k \in \mathbb{R}^k$ represent the columns of $A$. The least singular value of $A$ can be lower-bounded using the *leave-one-out* distance $\ell(A)$ which is defined as

$$\ell(A) = \min_i \text{dist}(a_i, V_{-i}), \tag{34}$$

where $V_{-i} = \text{span}\{a_j : j \neq i\}$ and $\text{dist}(x, V) = \min_{v \in V} \|x - v\|_2$ is the perpendicular $\ell_2$ distance between $x$ and the subspace $V$. The least singular value $s_k(A)$ is related to $\ell(A)$ by

$$\frac{\ell(A)}{\sqrt{k}} \leq s_k(A) \leq \ell(A). \tag{35}$$

We now lower-bound the leave-one-out distance $\ell(A)$. Fix a column $i \in [k]$. Let $u$ be any unit vector in the subspace orthogonal to $V_{-i} = \text{span}(\{a_j : j \in [k] \setminus \{i\}\})$; note that such a direction exists since $\dim(V_{-i}) \leq k - 1$. Moreover, since the column $a_i \sim N(\mu \mathbf{1}_k, I)$ where $\mathbf{1}_k = (1, 1, \ldots, 1)$ is the all-ones vector, we have $a_i^\top u \sim N(z, 1)$ where $z = \mu u^\top{}_k$. From the anti-concentration of Gaussian $N(z, 1)$, we have for an absolute constant $c > 0$,

$$\Pr\left[|u^\top a_i| \leq \delta\right] \leq \sup_{t \in \mathbb{R}} \Pr_{g \sim N(0,1)}\left[g \in [t - \delta, t + \delta]\right] \leq c\delta.$$

By picking $\delta = 1/(c(k+1)^9)$, we have with probability at least $1 - \frac{1}{(k+1)^9}$,

$$\text{dist}(a_i, V_{-i}) \geq |u^\top x| \geq \frac{1}{c(k+1)^9}.$$

By applying a union bound over all the columns $i \in [k]$, we have $\ell(A) \geq 1/(k+1)^9$ with probability at least $1 - \frac{1}{(k+1)^8}$. By applying (35), we see that part (i) of the lemma also holds.

$\square$

*Proof of Proposition 12.* Fix $i \in [k]$. The matrix $M[i]$ can be written in terms of the $(k-1) \times (k-1)$ submatrix $U^{(i)}$ as $M[i] = U^{(i)}(U^{(i)})^\top - \varepsilon I_{k-1}$. Each submatrix $U^{(i)}$ is a random matrix in $\mathbb{R}^{(k-1)\times(k-1)}$ with i.i.d. entries drawn from $N(\mu, 1)$ with $\mu > c_0\sqrt{\log k}$. By applying Lemma 12 with $k-1$ and choosing $c_0 > 2c_3$, we get that with probability at least $1 - 1/k^8$ that $\sigma_{k-1}(U^{(i)})^2 \geq 2c_1/k^{c_2}$. Hence,

$$\sigma_{k-1}(M[i]) = \sigma_{k-1}\left(U^{(i)}(U^{(i)})^\top\right) - \varepsilon \geq \sigma_{k-1}(U^{(i)})^2 - \frac{c_1}{k^{c_2}} \geq \frac{c_1}{k^{c_2}}.$$

Applying a union bound over all $i \in [k]$ proves the part (i) of the lemma.

Part (ii) follows since $U$ (and hence $UU^\top$) has nonnegative entries and $I$ only has diagonal entries. So all the off-diagonal entries of $M$ are nonnegative. The non-negativity of the diagonal entries follows from the positive semi-definiteness of the $M[i]$.

Finally part (iii) follows since $UU^\top$ is of rank $(k-1)$; hence $\lambda_k(M) = -\varepsilon$, which gives the required bound. $\square$

## E.2 Deterministic construction

We provide another construction of $k \times k$ strictly pseudo-PD matrices for any $k \geq 2$. Unlike the construction given in Appendix E.1, this construction is fully deterministic. However, it includes both positive and negative entries, making it an arguably less natural cost matrix for a weighted graph.

**Definition 5** (Almost-average matrix). *We define the $k \times k$ almost-average matrix $M$ as follows:*

$$M_{ij} = \begin{cases} 1 & \text{for } i = j, \\ -\frac{1}{k-1.5} & \text{for } i \neq j. \end{cases}$$

Now we show that $M$ is strictly pseudo-PD:

*Proof.* First we show that $M$ is not positive semidefinite. Consider a test vector $x$ of all 1's:

$$(Mx)_i = 1 - \frac{k-1}{k-1.5} = -\frac{0.5}{k-1.5}$$
$$Mx = -\frac{1}{2k-3} \cdot x$$

So $x$ is an eigenvector with a negative eigenvalue.

Now we show $M[i] \succeq 0$ for any $i \in [k]$. (Recall that $M[i] \in \mathbb{S}^{(k-1)\times(k-1)}$ denotes the submatrix of $M$ formed by deleting the $i$th row and column.) To see this, note that $M[i]$ is strictly diagonally dominant for any $i$ (for every row, the sum of the magnitudes of the non-diagonal entries is less than the magnitude of the diagonal entry). Any symmetric, strictly diagonally dominant matrix with nonnegative diagonal entries is positive definite. So $M[i]$ is indeed positive definite. $\square$

We additionally note that when the cost matrix takes the form (3) with $M$ defined as in Definition 5 (setting $k = n/2$), one can show that (MC-SDP) has a unique optimal rank-one solution. This implies in particular that if we view this instance of (MC-SDP) as a convex relaxation of a corresponding Max-Cut instance (albeit with perhaps unusual negative edge weights modeling "attractions"), the relaxation is tight. The following proposition will be used in Appendix G (Experiments).

**Proposition 13** (Optimal solution for cost matrix arising from almost-average matrix). *For any $n \geq 4$, consider the instance of* (MC-SDP) *with cost matrix*

$$A = \begin{bmatrix} M & M \\ M & M \end{bmatrix} \tag{36}$$

*where $M \in \mathbb{S}^{\frac{n}{2} \times \frac{n}{2}}$ is defined as in Definition 5 (setting $k = n/2$). The unique optimal solution of such an instance is the matrix $J \in \mathbb{S}^{n \times n}$ consisting of all $1$'s, implying in particular that the optimal objective value is*

$$4p \left( 1 - \frac{p-1}{p-1.5} \right),$$

*where $p$ is shorthand for $n/2$.*

Clearly one could extend Proposition 13 to the case where (36) is arbitrarily shifted by a diagonal matrix as in, e.g., Lemma 1, but we state it this way for simplicity and because we will use precisely matrices of the form (36) in Appendix G (Experiments). Proposition 13 is interesting because it provides an example of an instance of (MC-SDP) where $\widetilde{Y}\widetilde{Y}^\top$ and the unique globally optimal solution $J$ are qualitatively very different (e.g., one is rank $n/2$ and the other rank 1).

Before giving the proof of Proposition 13, we make the useful observation that *any* strictly pseudo-PSD (and therefore also strictly pseudo-PD) matrix has at most one negative eigenvalue.[9]

**Lemma 13.** *Let $B \in \mathbb{S}^{k \times k}$ be a strictly pseudo-PSD matrix for any $k \geq 2$. Then $B$ has exactly one negative eigenvalue.*

*Proof.* Any strictly pseudo-PSD matrix has at least one negative eigenvalue by definition. We show that if $B$ has more than one negative eigenvalue, one can construct an instance of (MC-BM) which contradicts a theorem due to [BVB18]. (It is the same theorem which ultimately yields the result that when $p > n/2$, (MC-BM) has no spurious second-order critical points.) Toward this goal, consider the instance of (MC-BM) with cost matrix

$$A = \begin{bmatrix} B & B \\ B & B \end{bmatrix} \in \mathbb{S}^{n \times n},$$

where we have defined $n := 2k$ for notational brevity. Due to Proposition 6, the axial position $\widetilde{Y}$ is a second-order critical point for this instance. Furthermore, it is easy to see that the multiplier $\nu \in \mathbb{R}^n$ associated with $\widetilde{Y}$ due to (1) is 0. Then Theorem 3.4 from [BVB18] gives that $A - \operatorname{diag}(\nu) = A$ has at most

$$\left\lfloor \frac{1}{k} \left( \dim \mathcal{F}_{\widetilde{Y}\widetilde{Y}^\top} - \frac{k(k+1)}{2} + n \right) \right\rfloor = \left\lfloor \frac{1}{k} \left( n - k \right) \right\rfloor = 1$$

negative eigenvalue. Here, $\mathcal{F}_{\widetilde{Y}\widetilde{Y}^\top}$ denotes the face of the convex feasible region of (MC-SDP) (also known as the elliptope) associated with $\widetilde{Y}\widetilde{Y}^\top$. (In other words, $\widetilde{Y}\widetilde{Y}^\top$ is in the relative interior of $\mathcal{F}_{\widetilde{Y}\widetilde{Y}^\top}$; see Definition 2.5 and Proposition 2.7 in [BVB18].) Above, we used the fact that $\dim \mathcal{F}_{\widetilde{Y}\widetilde{Y}^\top} = \frac{k(k+1)}{2} - k$ due to [BVB18, Prop. 2.7].

Next, note that

$$A = \begin{bmatrix} B & B \\ B & B \end{bmatrix} = \begin{bmatrix} 1 & 1 \\ 1 & 1 \end{bmatrix} \otimes B,$$

where $\otimes$ denotes the Kronecker product. Since the eigenvalues of $\begin{bmatrix} 1 & 1 \\ 1 & 1 \end{bmatrix}$ are $\{0, 2\}$, this decomposition implies that if $B$ had more than one negative eigenvalue, $A$ would have more than one negative eigenvalue, a contradiction. $\qquad \square$

---

[9]Whenever we use the phrases "at most one negative eigenvalue," "exactly one negative eigenvalue," etc., we are counting for multiplicity. So, for example, if a matrix has the negative eigenvalue -2 with multiplicity 4, this is counted as four separate negative eigenvalues (and, e.g., such a matrix could not be strictly pseudo-PSD due to Lemma 13).

Now we prove the main claim:

*Proof of Proposition 13.* $p$ will be used as shorthand for $n/2$ throughout this proof. Due to, e.g., [WW20, Prop. 1] or [BVB18, Prop. 2.8], $J$ is optimal if there exists some $\beta \in \mathbb{R}^n$ such that, defining $S := A - \text{diag}(\beta)$, we have $SJ = 0$ and $S \succeq 0$.[10] We claim these conditions hold when each entry of $\beta$ is set to $b := 2\left(1 - \frac{p-1}{p-1.5}\right)$. Indeed, $SJ = 0$ can be observed directly. As for $S \succeq 0$, note that

$$S = \left(\begin{bmatrix} 1 & 1 \\ 1 & 1 \end{bmatrix} \otimes M\right) - \text{diag}(\beta).$$

Since the spectrum of $\begin{bmatrix} 1 & 1 \\ 1 & 1 \end{bmatrix}$ is $\{0, 2\}$, the spectrum of $S$, denoted $\sigma(S)$, is precisely

$$\sigma(S) = \{-b\} \cup \{2\lambda - b : \lambda \in \sigma(M)\}. \tag{37}$$

Clearly $-b \geq 0$, so all that is left is to check that all elements of $\{2\lambda - b : \lambda \in \sigma(M)\}$ are nonnegative. Due to Lemma 13, $M$ has at most one negative eigenvalue. (Note that any strictly pseudo-PD matrix is strictly pseudo-PSD.) Furthermore, we found in the proof that $M$ is strictly pseudo-PD (directly below Definition 5) that the all 1's vector is an eigenvector of $M$ with eigenvalue $-\frac{0.5}{p-1.5}$. Thus, this is the single negative eigenvalue of $M$. Finally, note that

$$-\frac{1}{p-1.5} - b = -\frac{1}{p-1.5} - 2\left(1 - \frac{p-1}{p-1.5}\right) = 0,$$

so we conclude that $S \succeq 0$. Thus, we have established that $J$ is optimal.

As for the uniqueness of $J$, it is a classical result that $J$ is an extreme point of the feasible region of (MC-SDP) (aka the elliptope)—see for example Definition 2.6 and Proposition 2.7 in [BVB18] or Appendix F.1 in [WW20]. [WW20, Prop. 2] then gives that if strict complementary slackness holds, meaning $\text{rank}(S) = n - \text{rank}(J) = n - 1$, then $J$ is the unique optimal solution. Indeed, this follows due to (37), the fact that $M$ only has a single negative eigenvalue, and the fact that $-b > 0$ for $p \geq 2$. $\qquad\square$

## F   Extending construction of spurious local minimum to $p < n/2$

Lemma 14 allows us to extend our construction of a spurious local minimum for the $p = n/2$ case to $p < n/2$. Indeed, it implies that our construction of a spurious local minimum for the instance of (MC-BM) with associated feasible region $\mathcal{M}_{2p,p}$ yields a construction of a spurious local minimum for the instance of (MC-BM) with associated feasible region $\mathcal{M}_{n',p}$, for all $n' \geq 2p$. Thus, one can construct an instance of (MC-BM) with a spurious local minimum when $p < n/2$ using the construction for the instance of (MC-BM) associated with $\mathcal{M}_{2p,p}$.

We note that Lemma 14 also holds if you replace "spurious local minimum" with "spurious first-order critical point" or "spurious second-order critical point," although we do not prove it here. However, the intuition is clear: we embed a "bad instance" into the higher-dimensional space, and design the cost matrix so that the bad instance does not interact with the added dimensions.

**Lemma 14.** *Let $(n, p) \in \mathbb{N} \times \mathbb{N}$ be such that there exists an instance of* (MC-BM) *with cost matrix $A \in \mathbb{S}^{n \times n}$ and feasible point $Y \in \mathcal{M}_{n,p}$ such that $Y$ is a spurious local minimum. Then for all $n' \geq n$, there exists a cost matrix $A' \in \mathbb{S}^{n' \times n'}$ and a feasible point $Y' \in \mathcal{M}_{n',p}$ such that $Y'$ is a spurious local minimum for the instance of* (MC-BM) *with cost matrix $A' \in \mathbb{S}^{n' \times n'}$ and feasible region $\mathcal{M}_{n',p}$.*

*Proof.* We claim that the following construction works:

$$A' = \begin{bmatrix} A & 0 \\ 0 & 0 \end{bmatrix} \in \mathbb{S}^{n' \times n'}, \qquad Y' = \begin{bmatrix} Y \\ G \end{bmatrix} \in \mathcal{M}_{n',p},$$

---

[10]This comes from the fact that if these conditions are satisfied, the primal-dual pair $(J, \beta)$ satisfies the KKT conditions of (MC-SDP), implying optimality since (MC-SDP) is a convex program.

where $G \in \mathbb{R}^{(n'-n) \times p}$ is arbitrary except with the single restriction that all of its rows are unit vectors. Since $Y$ is a local minimum for the instance of (MC-BM) with cost matrix $A$ and feasible region $\mathcal{M}_{n,p}$, there exists $\epsilon > 0$ such that for $Z \in \mathcal{M}_{n,p}$ with $\|Z - Y\| < \epsilon$, we have $\langle A, YY^\top \rangle \leq \langle A, ZZ^\top \rangle$. Now let $Z' \in \mathcal{M}_{n',p}$ with $\|Z' - Y'\| < \epsilon$. Let $\bar{Z} \in \mathcal{M}_{n,p}$ denote the submatrix of $Z'$ consisting of the first $n$ rows. Then

$$\left\langle A', Y'Y'^\top \right\rangle = \left\langle A, YY^\top \right\rangle \leq \left\langle A, \bar{Z}\bar{Z}^\top \right\rangle = \left\langle A', Z'Z'^\top \right\rangle,$$

since $\|\bar{Z} - Y\| \leq \|Z' - Y'\| < \epsilon$. Thus, $Y'$ is a local minimum.

To see that $Y'$ is spurious, the spuriousness of $Y$ implies that there exists $V \in \mathcal{M}_{n,p}$ such that $\left\langle A, VV^\top \right\rangle < \left\langle A, YY^\top \right\rangle$. Then define $V' = \begin{bmatrix} V \\ G \end{bmatrix} \in \mathcal{M}_{n',p}$, where $G \in \mathbb{R}^{(n'-n) \times p}$ is once again some matrix whose rows are unit vectors. Then,

$$\left\langle A', V'V'^\top \right\rangle = \left\langle A, VV^\top \right\rangle < \left\langle A, YY^\top \right\rangle = \left\langle A', Y'Y'^\top \right\rangle.$$

$\square$

# G Experiments

In this section, we empirically evaluate our construction of a spurious local minimum for (MC-BM) in the setting where $p = n/2$ (the largest possible value of $p$ before we are guaranteed to have no spurious minima). Our experiments suggest that the spurious local minima we construct have surprisingly large basins of convergence. (In comparison, our theoretical results only guarantee the existence of *some* positive measure basin of convergence.)

Our code and data can be found at https://github.com/vaidehi8913/burer-monteiro. This repository also contains a link to a visualizer for the $p = 2$ and $p = 3$ settings.

**Instance generation.** For our experiments, we use the deterministic construction of pseudo-PD matrices given in Section E.2. In other words, the cost matrix always takes the form

$$A = \begin{bmatrix} M & M \\ M & M \end{bmatrix}$$

where $M \in \mathbb{S}^{\frac{n}{2} \times \frac{n}{2}}$ is defined as in Definition 5 (setting $k = n/2$). We run trials for $n = 4, 50, 200, 1000$, each time setting $p = n/2$. Recall that due to Lemma 1, the axial position $\widetilde{Y}$ is a spurious local minimum for such cost matrices. Furthermore, due to Proposition 13, we know precisely what the optimal value is for such instances, allowing us to distinguish whether the optimization algorithm converged to a spurious point or a global optimum.

**Optimization setup.** We use the standard trust-region solver with default settings from the MATLAB manifold optimization package Manopt [BMAS14]. It is a second-order method which uses the gradient of the objective function (which we provide) and an approximation of the Hessian of the objective function found using finite differences (this is done automatically by their implementation).

**Initialization.** In each trial, we sample the initialization point $Y^{(0)}$ from a neighborhood of the axial position $\widetilde{Y}$. We make use of the fact that $\mathcal{M}_p$ is a product manifold and measure the distance to initialization in a row-wise manner, making the following definition:

**Definition 6** ($\beta$-close)**.** *We say $Y, Y' \in \mathcal{M}_p$ are $\beta$-close if $\|Y_i - Y_i'\| \leq \beta$ for all $i \in [n]$, where $Y_i \in \mathbb{R}^p$ denotes the $i$th row of $Y$.*

For each trial, we choose a perturbation magnitude $\pi > 0$, which specifies how far from $\widetilde{Y}$ our initialization point will be. We then generate a perturbation matrix $\Delta \in \mathbb{R}^{n \times p}$. Each entry of $\Delta$ is drawn $\sim \mathcal{N}(0, \frac{1}{p})$ (a Gaussian distribution with variance $\frac{1}{p}$). This ensures that $\Delta$ has approximately unit-norm rows. Then our initial point is given by

$$Y^{(0)} = \text{row-normalize}\left(\widetilde{Y} + \pi\Delta\right),$$

where $\text{row-normalize} : \mathbb{R}^{n \times p} \to \mathbb{R}^{n \times p}$ normalizes each row of the input. This ensures that $Y^{(0)}$ is effectively sampled a constant distance away from $\widetilde{Y}$ on $\mathcal{M}_p$ in a uniformly chosen direction.

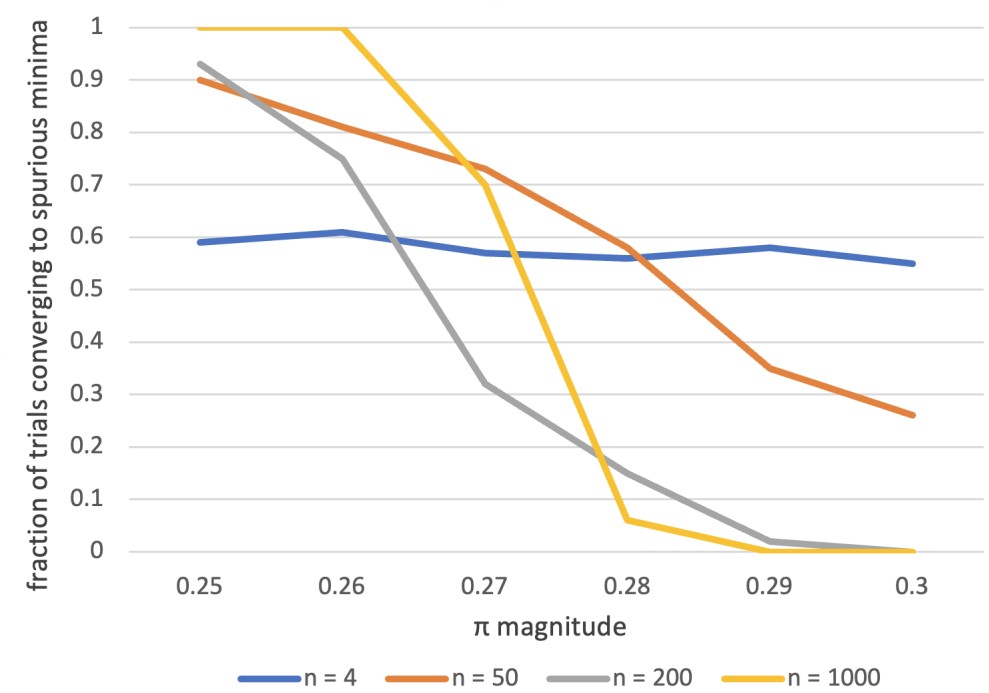

Figure 4: A plot summarizing our empirical results (data in Figure 5). Note that there appears to be a phase transition at $\pi \approx 0.27$, where trials almost always converge to spurious points for smaller $\pi$ and almost always converge to global optima for larger $\pi$. This phase transition appears to grow sharper with larger values of $n$.

**Data collection.** We report the fraction of trials in which the algorithm converged to a spurious point. We note that all trials we ran resulted in convergence to a point with objective value 0 (the objective value of our constructed spurious local minimum $\widetilde{Y}$) or to a point with the optimal (negative) objective value due to Proposition 13.

**Results.** We provide a summary of our experimental results in Figure 4 and the full data in Figure 5. For $n = 4$, we ran 1000 trials for each reported value of $\pi$. For $n = 50, 200$ we ran 100 trials for each $\pi$, and for $n = 1000$ we ran 50 trials for each reported value of $\pi$.

We note an interesting phase transition that seems to occur at $\pi \approx 0.27$. For perturbations greater than this threshold, the algorithm seems to almost always converge to a global optimum. Below this threshold, the algorithm seems to almost always converge to a spurious point. This threshold appears to get sharper as $n$ gets larger. This suggests there is some $\beta^* \approx 0.27$ such that points $\beta^*$-close to $\widetilde{Y}$ are very likely to converge to a spurious point, and vice versa. (It is also interesting that this family of cost matrices seems to have this phase transition at the same value for every $n$.) All in all, our experiments suggest a much larger basin of convergence for spurious minima than our theoretical results guarantee.

| $\pi$ | $n = 4,$ | 50, | 200, | 1000 |
|------|------|------|------|------|
| 0.3  | 0.55 | 0.26 | 0    | 0    |
| 0.29 | 0.58 | 0.35 | 0.02 | 0    |
| 0.28 | 0.56 | 0.58 | 0.15 | 0.06 |
| 0.27 | 0.57 | 0.73 | 0.32 | 0.70 |
| 0.26 | 0.61 | 0.81 | 0.75 | 1    |
| 0.25 | 0.59 | 0.90 | 0.93 | 1    |

Figure 5: *(Complete Data)* For each combination of $n, \pi$, we report the fraction of our trials that converged to spurious points. For $n = 4$, we ran 1000 trials for each reported value of $\pi$. For $n = 50, 200$, we ran 100 trials for each reported value of $\pi$. For $n = 1000$, we ran 50 trials for each reported value of $\pi$.