# OpenReview forum: "The Burer-Monteiro SDP method can fail even above the Barvinok-Pataki bound"
_NeurIPS.cc/2022/Conference — NeurIPS 2022 Accept_

### Official Review · Reviewer_tXww · 2022-07-09

**Rating:** 8
**Confidence:** 4
**Soundness:** 4 excellent
**Presentation:** 4 excellent
**Contribution:** 3 good

**Summary:**

Large SDPs are costly to solve via interior point methods. Therefore, there is great interest in techniques that are much faster and achieve nearly optimal results. Burer-Monteiro (BM) is one such method: it seeks a SDP solution -- a $n \times  n$ matrix $X$ -- which has rank $k$, and parameterizes $X=V^TV$, where $V$ is $k \times n$. $V$ is then optimized, and the SDP constraints then become restrictions on the rows of V.

The present paper studies the BM method for the well-known SDP relaxation of MaxCut. Specifically, the main question is whether local maxima of BM correspond to global maxima of the SDP. The focus is on the regime $k(k+1)/2>n$. In this regime, there exist rank k solutions that are optimal for the (full rank) SDP. However, standard BM is only guaranteed to find local maxima. The question then becomes: is it always the case that a local maximum for BM is a global maximum also (and thus a global optimum for the SDP)?

Quite simply, this paper shows that the answer is *no* in general, for a large range of $k$ up to $n/2$. This is shown by an explicit construction, and nicely complements a result by Boumal, Voroninski and Bandeira that showed that the answer is *yes* for "generic" instances of the MaxCut SDP. The present paper also shows that there " bad"  local maxima are stable for gradient ascent.


**Questions:**

This is an optional question, but can any of the SO(d) synchronization tasks in the paper by Mei et al be similarly analyzed (ie. there are "natural" instances where SDP and local solutions of the low-rank BM are far apart)?

**Limitations:**

See the weakness above. My only comment is that the contribution by Mei et al should be discussed.

**Strengths And Weaknesses:**

The main strength of the present paper is that it in some sense finishes the "landscape analysis" of BM for the MaxCut SDP. This is an interesting result that has strong connections with important recent work. The construction in the proof is nice and original. The analysis of how the objective function behaves around the "bad local minimum" is also interesting, and uses the machinery of Riemannian optimization as well as probabilistic arguments.

The main weakness is that in some sense we expect local optimization methods to do quite well, as second-order local minima of Burer-Monteiro are nearly optimal for the SDP. This is shown in:

Mei et al. "Solving SDPs for synchronization and MaxCut problems via the Grothendieck inequality," COLT 2017.

Indeed, a rank-k local maximum achieves the SDP maximum up to a (1-O(1/k)) error, whenever the SDP instance comes from a weighted graph (with nonnegative weights on the edges). For $k\geq \sqrt{2n}$ (as considered here) this means that the rank-k solutions are close to the full rank ones. Since MaxCut involves a rounding with an approximation factor anyway, one could argue that the (small) extra error from Met et al is not very substantial.

Still, to be clear, I think this is a strong paper, that in a way finishes an important line of research on how "benign" BM is as a SDP solution method.

---

> ### Author Response · Authors · 2022-07-31
> **Applications of the maxcut SDP beyond the maxcut problem itself**
>
> *“The main weakness is that in some sense we expect local optimization methods to do quite well, as second-order local minima of Burer-Monteiro are nearly optimal for the SDP. This is shown in:
> Mei et al. "Solving SDPs for synchronization and MaxCut problems via the Grothendieck inequality," COLT 2017.
> Indeed, a rank-k local maximum achieves the SDP maximum up to a (1-O(1/k)) error, whenever the SDP instance comes from a weighted graph (with nonnegative weights on the edges). For $k \ge \sqrt{2n}$ (as considered here) this means that the rank-k solutions are close to the full rank ones. Since MaxCut involves a rounding with an approximation factor anyway, one could argue that the (small) extra error from Met et al is not very substantial.”*
>
> Thank you for pointing out this paper by Mei et al that shows that for instances of this SDP that correspond to valid max-cut instances, we can expect the local and global minima for (BM) to be quite close in solution value up to a small multiplicative (approximation) factor.  We will certainly incorporate it into our discussion.
>
> However, while the max-cut SDP is of course used to solve max-cut, the name is in some ways a misnomer, as this form of SDP can be used to solve a large number of problems.  For example, this SDP is used in the convex relaxation of quadratic programming, community detection problems, the Grothendieck problem etc. The proof in Mei et al. is quite specific to max-cut and does not extend to these more general problems.   In fact it is unlikely that such a constant multiplicative gap can be shown for local minima solution value to the globally optimum solution value (for e.g., quadratic programming we expect this factor to be larger than logn when rank=1).
>
> Moreover, there are several applications like community detection, where the solution value is not as important as the optimum SDP solution itself (which is used to recover some ground-truth solution).  In a broad sense, we feel that the main contribution of our paper is not to analyze a particular way to solve the max-cut problem.  Rather, before our work it was not known whether there was any SDP, maxcut or otherwise, for which the Burer-Monteiro method could fail above the Barvinok-Pataki bound.  One could have reasonably conjectured that the Burer-Monteiro method always worked.  Our work shows a counterexample, proving this statement false.  In some ways, the maxcut SDP is incidental, we only needed one counterexample.
>
> However, it is worth noting that max-cut (or quadratic programming) is a basic SDP, which is why many works start by first considering it.  Previous to our work, it could have been reasonable to guess that such a nicely behaved function class would not have spurious local minima.  We think it is noteworthy and surprising that there are spurious minima, especially for such a basic SDP, and that too at such high rank.
>
> *“This is an optional question, but can any of the SO(d) synchronization tasks in the paper by Mei et al be similarly analyzed (ie. there are "natural" instances where SDP and local solutions of the low-rank BM are far apart)?”*
>
> This is an interesting question. We have not thought about trying to extend our construction (or tried to come up with a new construction) for SO(d) when d > 1. In terms of extending our results to other SDPs, this is clearly a very natural direction for future work.

---

> > ### Comment · Reviewer_tXww · 2022-08-08
> > **Thank you!**
> >
> > Thank you for your comments. I remain convinced that this is a "strong accept".

---

### Official Review · Reviewer_4PkM · 2022-07-11

**Rating:** 7
**Confidence:** 3
**Soundness:** 4 excellent
**Presentation:** 4 excellent
**Contribution:** 3 good

**Summary:**

This paper studies the landscape of the Burer-Monteiro optimization problem corresponding to the classical Goemans-Williamson SDP relaxation of Max-Cut. It was known that for generic edge weights, the Burer-Monteiro optimization over n x p matrices has no spurious second-order critical points once the rank constraint exceeds the Barvinok-Pataki bound, p(p+1)/2 >= n. However, it was unknown whether there exist particular edge weight matrices for which spurious second-order critical points or local maximizers can exist.

This work provides an explicit construction of a class of (signed) edge weight matrices, for which the BM-optimization of size 2p x p has a spurious local maximizer. By zero-padding, this construction extends also to the BM-optimization of size n x p for any n >= 2p. The construction is to posit a specific matrix Y = [I; -I] in R^{2p x p} as the candidate for the spurious local maximizer, and to then characterize those edge weight matrices for which this Y is a first-order critical point, second-order critical point, and spurious local maximizer, respectively. The characterizations of first-order and second-order criticality are simpler and based upon explicit forms of the Riemannian gradient and Hessian. However, as the Hessian is rank-deficient for p = n/2 and there is an entire sub-manifold of local minimizers containing Y, a more involved geometric argument that explicitly shows convergence of Riemannian gradient descent to this submanifold is used to argue that Y is a local minimizer. Experiments are provided in the appendix that explore the basin of attraction of this local minimizer.

**Questions:**

(1) The author(s) describe the non-negativity of the probabilistic construction of the cost matrix as an advantage---but since the problem has been formulated with min <A,X> rather than max <A,X>, wouldn't having all negative (rather than positive) entries correspond more naturally to instances of Max-Cut on graphs? Is it known whether there exists a matrix A with all negative entries, for which BM has a spurious local minimizer?

(2) The detailed discussions of the proofs of Lemmas 4 and 5 on lines 287-308 are a bit hard to follow, and I don't know if they add much value to the paper. I do find some of the results on the basins of attraction in Appendix F to be interesting, and it might be nice to move some portion of this to the main text. One question here is---for the 75% of trials that converge to spurious minimizers, are they converging to antipodal points, or are they discovering other types of local minima?

Typos:

l173, M should be P

p19 starts mixing notation w_{ij} and M_{i,j} for the entries of M

l623, the following display is missing a factor p

l684, the following display is missing some constant factors

l906, should this be "P contained in *kernel* of Hess Obj(tilde{Y})"?

l934, rank k should be k-1

l937, U in R^{k x k} should be {k x k-1}

l942, k should be k-1, and l944, k+1 should be k

l964, sigma_k should be s_k

l1012 and the subsequent display, should > be <?

**Limitations:**

Yes

**Strengths And Weaknesses:**

I think this is an interesting paper, and would be supportive of its publication in NeurIPS.

The result is a bit limited in scope, in that it studies only the specific Goemans-Williamson SDP, and constructs only a specific type of local optimum for BM-optimization of this SDP. However, I find the identification of this particular type of local optimizer Y (the axial position matrix) to be insightful, and it is nice that the construction works all the way to the sharp threshold p = n/2. The analysis to show that it is a local optimizer, rather than just a second-order critical point, is also non-trivial due to the rank degeneracy of the Hessian, and I think it develops an interesting proof idea. Overall, I think the paper is insightful, well-written, and resolves a question whose answer was previously unknown in this literature.

---

> ### Author Response · Authors · 2022-07-31
> **Wide applicability of maxcut SDP, trials do not seem to always converge to antipodal points**
>
> Thank you for catching these typos, we will incorporate them in our edits.
>
> *“The result is a bit limited in scope, in that it studies only the specific Goemans-Williamson SDP, and constructs only a specific type of local optimum for BM-optimization of this SDP.”*
>
> Before our work it was not known whether there was any SDP, maxcut or otherwise, for which the Burer-Monteiro could fail above the Barvinok-Pataki bound.  One could have reasonably conjectured that the Burer-Monteiro method always worked.  Our work shows a counterexample, proving this statement false.  In some ways, the maxcut SDP is incidental, we only needed one counterexample.
>
> However, it is worth noting that while this SDP is often called the Goemans-Williamson maxcut SDP, it is a classic SDP that is also used for many other problems, including the convex relaxation for general quadratic programming, community detection, the Grothendieck norm problem etc., and many other combinatorial optimization and machine learning problems (e.g., this SDP is also used in certifying adversarial robustness) It is arguably the most popular SDP and very simple, which is why many works start by first considering the maxcut SDP.  Previous to our work, it could have been reasonable to guess that such a nicely-behaved function class would not have spurious local minima.  We think it is noteworthy and surprising that there are spurious minima, especially for such a widely used SDP, and that too at such high rank.
>
> *1. “The author(s) describe the non-negativity of the probabilistic construction of the cost matrix as an advantage---but since the problem has been formulated with min <A,X> rather than max <A,X>, wouldn't having all negative (rather than positive) entries correspond more naturally to instances of Max-Cut on graphs? Is it known whether there exists a matrix A with all negative entries, for which BM has a spurious local minimizer?”*
>
> Perhaps counterintuitively, it is actually the minimization problem that corresponds to max-cut.  One way to think about this is that the max-cut SDP objective function measures the “closeness” of the vertices.  The SDP then tries to minimize this “closeness.”  This corresponds to max-cut, which tries to maximize the distance between vertices.  There are other SDP formulations that maximize a function of the graph Laplacian matrix (rather than the adjacency matrix) but this is not what we consider here.
>
> However, it is worth noting that while this SDP is often called the max-cut SDP, it is used far more broadly than for just max-cut.  For example, this form of SDP is also used in the convex relaxation of quadratic programming.  So looking at cost matrices that are not necessarily nonnegative is well motivated.
>
> To your question about whether there is a matrix with all nonnegative entries for which (BM) has a spurious local minimum, we do not know.  Our construction would not work for such a matrix, but there may be some other construction that would.
>
> *2. “The detailed discussions of the proofs of Lemmas 4 and 5 on lines 287-308 are a bit hard to follow, and I don't know if they add much value to the paper. I do find some of the results on the basins of attraction in Appendix F to be interesting, and it might be nice to move some portion of this to the main text. One question here is---for the 75% of trials that converge to spurious minimizers, are they converging to antipodal points, or are they discovering other types of local minima?”*
>
> We appreciate the feedback about clarity and will consider this in our edits.
>
> Thank you for the question about whether the trials corresponding to spurious minimizers are converging to antipodal points, this is a very interesting question. Some preliminary (non-rigorous) trials on small instances indicate that they are indeed discovering many different types of local minima that are not necessarily antipodal. This does not contradict our theoretical result, that says that these types of instances have at least one spurious local minimum. This would be an interesting direction for further investigation.

---

> > ### Comment · Reviewer_4PkM · 2022-08-07
> > **Thanks for clarification**
> >
> > Thanks for the clarification regarding positivity/negativity of A---you're right, and I was confusing this with similar relaxations of minimum bisection.

---

### Official Review · Reviewer_sULy · 2022-07-12

**Rating:** 8
**Confidence:** 3
**Soundness:** 3 good
**Presentation:** 3 good
**Contribution:** 4 excellent

**Summary:**

This work studies the Burer-Monteiro method, a nonconvex method for solving semidefinite programs (SDPs) that have only equality constraints and whose solution is a $n \times n$ PSD matrix of rank $p$. The solution to the Burer-Monteiro method is known to coincide with the original SDP when the rank is above the Barvinok-Pataki bound $p \gtrsim \sqrt{2n}$. Although the Burer-Monteiro problem is noncovex, it is known to be solvable in polynomial time in a smoothed analysis setting when $p \gtrsim \sqrt{2n}$, and it is known to be solvable in polynomial time for worst-case instances when $p > n/2$. The main result of this paper constructs worst-case instances for the max-cut SDP for all ranks $\sqrt{2n} \lesssim p \leq n/2$ whose corresponding Burer-Monteiro problem has spurious local minima, and hence gradient-descent type approaches fail to solve this problem. This suggests that the use of beyond-worst case analysis frameworks is necessary for proving the efficiency of the Burer-Monteiro method.

**Questions:**

**1.** The method for proving existence of local minima uses Riemannian gradient descent. Are there other works that use a similar technique?

**Limitations:**

**1.** The authors do not foresee negative societal impacts, and I agree with this assessment.



**Strengths And Weaknesses:**

# Strengths

- *Significance*: The Burer-Monteiro has attracted interest in recent years, and it has remained open whether or not a smoothed analysis setting was necessary for polynomial time algorithms above the Barvinok-Pataki bound.

- *Quality/originality*: The proof of existence of local minima relies on a notion of (strictly) pseudo-PSD matrices and a careful analysis of Riemannian gradient descent. In my view these ideas are interesting and the technical level is moderately high.

- *Clarity*: The paper is well-written and organized.

# Weaknesses

- I do not see any major weaknesses. I just have a quibble in the next bullet about how the contribution of this paper is phrased.

- *Clarity*: I find the phrase "The Burer-Monteiro method fails..." to be confusing. It's not clear what it means to "fail" since solving the Burer-Monteiro optimization problem solves the original SDP in a certain regime. The real problem is that the Burer-Monteiro solution may not be efficiently computable. This paper gives evidence of computational hardness by demonstrating the presence of spurious local minima in worst-case instances, but it does not strictly rule out some other clever polynomial time algorithm from solving the max-cut Burer-Monteiro problem.

# Minor comments

**1.** I think it would improve the clarity of this paper to first introduce the Burer-Monteiro method in general (ie for general SDPs with linear equality constraints and rank p solutions) and then specialize to the max-cut problem.

**2.** The preliminaries are introduced abruptly, and if the paper is read linearly, it is not clear initially why some of them are needed. Either (i) a high-level sketch of the proof of Theorem 1 before the preliminary section or (ii) a short sentence for each preliminary describing why it is needed (eg: we use Riemannian gradient descent to prove existence of local minima) could help with this.

**3.** The notation $ p \gtrsim \sqrt{2n}$ might be confused with $ p = \Omega( \sqrt{2 n} )$, although the latter is not what is meant here.

---

> ### Author Response · Authors · 2022-07-31
> **Title explanation and novel gradient descent method**
>
> Thank you for your comments regarding clarity and readability. We will consider them in our edits.
>
> *“I find the phrase "The Burer-Monteiro method fails..." to be confusing. It's not clear what it means to "fail" since solving the Burer-Monteiro optimization problem solves the original SDP in a certain regime. The real problem is that the Burer-Monteiro solution may not be efficiently computable. This paper gives evidence of computational hardness by demonstrating the presence of spurious local minima in worst-case instances, but it does not strictly rule out some other clever polynomial time algorithm from solving the max-cut Burer-Monteiro problem.”*
>
> When we refer to the “Burer-Monteiro method” or heuristic, we mean a specific approach to solving the original SDP optimization problem by running local algorithms (like Riemannian gradient descent) on the rank-constrained non-convex objective (this is consistent with the terminology and phrasing of prior works in this space). The Burer-Monteiro method is a popular heuristic in practice, and prior works have tried to explain the success of this specific heuristic on generic or smoothed instances when the desired rank is above the Barvinok-Pataki bound (by showing that all local minima are non-spurious). Since we establish that there are bad instances with spurious local minima, it shows that the Burer-Monteiro heuristic doesn’t always work even above the Barvinok-Pataki bound.
>
> The rank-constrained non-convex problem (that the reviewer refers to as the Burer-Monteiro problem) has the same global optimum solution, since the desired rank is above the Barvinok-Pataki bound. And as the reviewer pointed out, our paper does not rule out a polynomial time algorithm for this non-convex optimization problem (but this is different from the Burer-Monteiro method) – in fact it cannot, since we know other polynomial time algorithms in this regime (solve the SDP using interior point method/Ellipsoid method, and then use the Barvinok-Pataki procedure). However our result proves that the Burer-Monteiro method/ heuristic fails. We will clarify this point in the paper.
>
> *“The method for proving existence of local minima uses Riemannian gradient descent. Are there other works that use a similar technique?”*
>
> We thank the reviewer for bringing up this point. We are not aware of any other works that use a similar technique.  In some previous works, the authors directly argue that the Hessian is positive definite at a certain point to show it is a local minimum.  However, in our problem, the Hessian is necessarily rank deficient, which inspired us to develop this (to the best of our knowledge) novel technique. We will try to emphasize this more in the final version.

---

> > ### Comment · Reviewer_sULy · 2022-08-08
> > **Reply to authors**
> >
> > Thanks very much for the helpful reply. I see your point and of course agree with you now that Burer-Monteiro *is* actually efficiently computable via SDP. If the usage of "BM fails" here is consistent across the literature and also you expand upon what "BM fails" really means in the revision, then this resolves my quibble. Thank you.

---

### Official Review · Reviewer_nF9P · 2022-07-13

**Rating:** 3
**Confidence:** 2
**Soundness:** 3 good
**Presentation:** 3 good
**Contribution:** 1 poor

**Summary:**

**Background for the paper**

Semidefinite programs (SDPs) form a popular class of convex programs but suffer from large input size in most useful settings. To circumvent this issue of size, Burer and Monteiro proposed expressing the problem variable as
$X=YY^{\top},$ for some $Y\in\mathbb{R}^{n\times p}$ and, instead of (SDP), solving the following proxy problem, which we refer to as (BM):
\begin{align*}
\min_{Y\in\mathbb{R}^{n\times p}, YY^\top \in\mathcal{C}}C\bullet YY^{\top},
\end{align*}
where $\mathcal{C}$ is the constraint set for the original SDP with $m$ constraints. The benefit conferred by
this reduction is that (BM) uses $O(np)$ memory, whereas (SDP) uses $O(n^2)$.

(BM) clearly lies in a lower rank space than (SDP), and therefore it shouldn't necessarily be the case that solving (BM) solves (SDP). However, it turns out that when $p$ (in the size of $Y$ above) satisfies $p  = O(\sqrt{m})$, then the solution set of (BM) also contains that o (SDP), following the celebrated result of Barvinok and Pataki (independently obtained) that there
exists a rank-$\sqrt{m}$ solution to (SDP).

In opposition to the memory saving, an immediate disadvantage of the formulation of BM is its non-convexity. Standard convex optimization techniques therefore fail to provide convergence guarantees. There has been a flurry of recent work (Boumal-Voroninski-Bandeira, Bhojanapalli-Boumal-Jain-Netrapalli, Cifuentes, and Cifuentes-Moitra) that provide polynomial-time guarantees for the BM formulations of various classes of SDPs, *in the smoothed analysis setting*.

**What problem this paper studies**

In the line of work mentioned above, ``smoothed analysis'' is central to the theorem statements. In other words, all the guarantees provided by those papers exclude a set of cost matrices. (The reason for this is that those papers prove their guarantees by applying a small perturbation to the cost matrices and showing that the probability of the algorithm ending up in a set of spurious critical points --- matrices that are (second-order) stationary points for BM but do not correspond to solutions to the original SDP --- is vanishingly small.) **The question posed by this paper is if such an exclusion is necessary**.

The paper then provides an affirmative answer to this question by an explicit construction of such an SDP.  It uses tools from Riemannian calculus to substantiate a point being a second-order stationary point while its corresponding point being suboptimal for the original SDP.



**Questions:**

Please see above.

**Limitations:**

Please see above.

**Strengths And Weaknesses:**

Based on my current understanding of the paper's contribution, my preliminary review is rejection. This is because my understanding is that this problem has been solved by Bhojanapalli-Boumal-Jain-Netrapalli (http://proceedings.mlr.press/v75/bhojanapalli18a/bhojanapalli18a.pdf). See the paragraphs after Corollary 4 and also the formal statement in Theorem 5.

Could the authors please explain to me how their result differs from this one? It's highly likely that I'm misunderstanding the contribution, in which case I want to learn what I'm missing (and I'll of course update my score accordingly!). I look forward to reading the authors' response.

----------------------------------------------------------------------------------------------------------------------------------------------------

After the rebuttal: it's clear that I am missing the key contribution here. I am going to try to figure this out but, for now, will change my confidence score since I am no longer confident of my original assessment. I hope I can better understand this work's contribution, but if that doesn't happen soon enough, I'll request the AC to ignore my input.

---

> ### Author Response · Authors · 2022-07-31
> **Bhojanapalli-Boumal-Jain-Netrapalli analyzes a different optimization problem**
>
> *“This is because my understanding is that this problem has been solved by Bhojanapalli-Boumal-Jain-Netrapalli (http://proceedings.mlr.press/v75/bhojanapalli18a/bhojanapalli18a.pdf). See the paragraphs after Corollary 4 and also the formal statement in Theorem 5.”*
>
> The Bhojanapalli-Boumal-Jain-Netrapalli paper actually analyzes a different optimization problem than (BM).  They formulate an unconstrained “penalized” optimization problem, in which the original constraints of (BM) are wrapped into a “penalty” term in the objective function.  This is different from the (BM) optimization problem, which optimizes a simpler objective function, while being constrained to a manifold.
>
> Their construction of a spurious critical point for this penalized version of the problem is not a spurious critical point for (BM). In their construction the cost matrix $C$ is the zero matrix, which will obviously not work in any construction for (BM) for exhibiting spurious critical points.  Furthermore, their construction involves creating constraints that exploit the nonconvexity of the penalty term in the objective function.  However this penalty term does not exist in standard (BM), so we cannot expect this construction strategy to work for standard (BM).
>
> For many SDPs solved in practice e.g., the maxcut SDP finding some feasible point and maintaining feasibility is simple.  Because of this, the standard formulation of (BM) is commonly used, and this is the version that we study, along with many other previous works (see for example Boumal-Voroninski-Bandeira https://arxiv.org/pdf/1804.02008.pdf).  As an aside, the main downside of the penalty formulation of Bhojanapalli-Boumal-Jain-Netrapalli is that you may need to set the penalty parameter $\mu$ from Equation (4) to be very large to ensure approximate feasibility, at which point there may be ill conditioning. The authors acknowledge this in the second paragraph of the conclusion in the arXiv version of their paper (https://arxiv.org/abs/1803.00186). Also, we remark that one of the authors of the Bhojanapalli-Boumal-Jain-Netrapalli-2018 paper is also an author of the Boumal-Voroninski-Bandeira-2018 paper -  ArXiv revision in 2019, which states our main result as being open (see end of page 22).
>
> We appreciate you pointing out the connection to the Bhojanapalli et al work.  We will incorporate a comparison to their result into our paper.

---

> > ### Author Response · Authors · 2022-08-08
> > **Any further clarification or follow-up question?**
> >
> > We thank you again for your thoughtful review and comments. We hope that in our author response, we have addressed your main concern/ misunderstanding about the comparison to BBJN. We will clarify this in more detail in the final version. With the author-reviewer period ending soon, we just wanted to check in and see if there are any follow up questions or other clarification questions you may have. Thank you!

---

> > > ### Comment · Reviewer_nF9P · 2022-08-09
> > > **Thank you for your clarification!**
> > >
> > > Dear authors,
> > >
> > > Thank you for your detailed rebuttal. I apologize for the radio silence on my end.
> > >
> > > I had been trying to understand your key idea on my own (and had also asked the other reviewers and AC privately) so as to be able to better review your work.
> > >
> > > I do not have any further questions for you but I'll make sure I engage with the AC and other reviewers in the discussion phase to improve my understanding of your contribution and, hopefully, your score as well.
> > >
> > > If I make no headway into understanding this (as of now, it's clear that I seem to be missing the point), then I'll request the AC to simply ignore my input and consider only the other (clearly far better) reviews.
> > >
> > > Thank you again for your patience, and congratulations on the great scores from the other reviewers :)

---

> > > > ### Author Response · Authors · 2022-08-09
> > > > **Further clarification**
> > > >
> > > > Thank you for getting back to us – we appreciate it very much.
> > > >
> > > > Perhaps one point that may help clarify the challenge with lower bounds for standard Burer-Monteiro, as opposed to the penalized version in BBJN: the BBJN bad instance sets $C=0$, and *only* considers the non-convex objective that arises from penalizing the constraints. Hence, their construction primarily concerns whether the penalized version can find a feasible solution for the program – it’s not concerned with the objective since $C=0$. Their construction is also not for max-cut, but for a problem with more complicated constraints.
> > > >
> > > > In the case of the Max-Cut SDP, this feasibility question is just about solving the optimization problem $\mu \sum_i ( \| Y_i\|^2 -1 )^2 $. It turns out that this problem actually has no spurious minima! This shouldn’t be surprising since we can always get a feasible solution by scaling all of the vectors to be unit vectors. So, this tells us why we can’t expect their lower bound construction to be useful for arguing about the standard Burer-Monteiro formulation, even in the case of the max-cut SDP. To reemphasize, their penalized formulation is an entirely different optimization problem than (BM), and the instance of (BM) associated with the constraints and objective matrix of their spurious critical point construction for the penalized formulation trivially does not have spurious local minima or spurious critical points since $C = 0$.
> > > >
> > > > In terms of our proof strategy, our starting point is the realization that a structured solution $\widetilde{Y}$ can be a spurious critical point if the objective matrix satisfies the strictly pseudo-positive-definite condition that we identify. The instance is already interesting when $p=2$ (and $n=4$), and the argument is non-trivial even in this case, and carries a nice geometric intuition. The most challenging/technical part of the argument is to prove that this is indeed a local minima. One issue is that there are many symmetries in the problem e.g., if $Y$ is a solution, so is $YO$ where $O$ is any orthogonal matrix (since $YY^T = YOO^T Y^T$. So the objective is not strictly convex around $\widetilde{Y}$;  in fact, even looking at higher order derivatives doesn’t help due to degeneracy (see Section C.4). Our proof instead introduces a surrogate objective/potential around $\widetilde{Y}$, and argues that Riemannian gradient descent starting from any point in a basin around $\widetilde{Y}$ converges to a point with potential value = 0 (this may not be $\widetilde{Y}$, but it has the same objective value as $\widetilde{Y}$).

---

### Meta-Review · Area_Chair_p2jR · 2022-08-25

**Recommendation:** Accept
**Confidence:** Certain

**Metareview:**

The Burer-Monteiro method is widely used for solving large scale semidefinite programs. It works based on replacing an $n \times n$ positive semidefinite matrix $X$ with $Y Y^T$ where $Y$ is $n \times p$. This has the benefit that it is more space efficient to store $Y$ than $X$, but it transforms a convex optimization problem into a nonconvex one. Above the Barvinok-Pataki bound (an analogue of the notion of a basic feasible solution for semidefinite rather than linear programming) we are at least guaranteed that there is a low-rank optimal solution. But does the nonconvex problem have spurious critical points? Recent works have studied the critical points under a smoothed analysis model, and shown that the Burer-Monteiro method works almost down to the Barvinok-Pataki bound. The main contribution of this paper is to complete the analysis of the landscape, by showing that without smoothing, even for the MAX-CUT SDP, there are spurious critical points even for $p = n/2$. One reviewer had doubts about the relationship to the work of Bhojanapalli-Boumal-Jain-Netrapalli, but I found the author reply to be convincing that the setting and techniques are fundamentally different. The other reviewers were uniformly positive. This is a nice contribution to the literature on the Burer-Monteiro method.

As a comment to the authors, I would suggest elaborating on the connection to the work of Mei et al. I agree that showing the global and local optima are close in objective value can be somewhat orthogonal to showing that the SDP recovers e.g. some underlying clustering in community detection. This provides a further justification why it is important to understand the loss landscape, and not just bound the suboptimality of any locally optimal solution. Indeed, from what I remember of the Mei et al. paper, the locally optimal solutions do not get non-trivial performance for the associated community detection problem, so I think investigating this further and explaining it would be helpful, since these are subtle distinctions.

**Award:**

No

---

### Decision · Program_Chairs · 2022-09-14

Accept